# Phylogenetic divergence of GABAB receptor signaling in neocortical networks over adult life

Max A. Wilson [1,2,3], Anna Sumera[1,2,3], Lewis W. Taylor[1,4], Soraya Meftah [1,4], Robert I. McGeachan [1,4], Tamara Modebadze[5], B. Ashan P. Jayasekera[5], Christopher J. A. Cowie[6], Fiona E. N. LeBeau[5], Imran Liaquat[7], Claire S. Durrant [1,4], Paul M. Brennan [7,8] & Sam A. Booker [1,2,3] ✉

Cortical circuit activity is controlled by GABA-mediated inhibition in a spatiotemporally restricted manner. GABAB receptor (GABABR) signalling exerts powerful slow inhibition that controls synaptic, dendritic and neuronal activity. But, how GABABRs contribute to circuit-level inhibition over the lifespan of rodents and humans is poorly understood. In this study, we quantitatively determined the functional contribution of GABABR signalling to pre- and postsynaptic domains in rat and human cortical principal cells. We find that postsynaptic GABABR differentially control pyramidal cell activity within the cortical column as a function of age in rodents, but minimally change over adult life in humans. Presynaptic GABABRs exert stronger inhibition in humans than rodents. Pre- and postsynaptic GABABRs contribute to co-ordination of local information processing in a layer- and species-dependent manner. Finally, we show that GABABR signalling is elevated in patients that have received the anti-seizure medication Levetiracetam. These data directly increase our knowledge of translationally relevant local circuit dynamics, with direct impact on understanding the role of GABABRs in the treatment of seizure disorders.

The transmission of information within cortical circuits requires integration of synaptic inputs to precisely control action potential output, which is true of all mammals. In mature neurons, this flow of excitatory information is largely orchestrated by the activation of GABA receptors which reduce neuronal activity by providing hyperpolarising or shunting inhibition to neuronal membranes in a cell-compartment-specific manner[1]. While much is known about GABAARs, determining how GABABRs contribute to neuronal function across brain development is critical, as they are implicated in the aetiology of seizures and epilepsy in humans[2–4] and preclinical rodent models[5–7].

The mammalian neocortex displays both a columnar and laminar structure, with six defined layers extending from superficial layer 1 (L1) to deep L6. Layers 2–6 are populated with principal cells (PCs) which possess dendrites that terminate locally and in L1[8]. While cortical columns and lamina organisation are generally common to all mammals, increased human cortical thickness allows greater subdivision of

[1]Centre for Discovery Brain Sciences, University of Edinburgh, Edinburgh EH8 9XD, UK. [2]Simons Initiative for the Developing Brain, University of Edinburgh, Edinburgh EH8 9XD, UK. [3]Patrick Wild Centre, University of Edinburgh, Edinburgh EH8 9XD, UK. [4]UK Dementia Research Institute at the University of Edinburgh, Edinburgh EH16 4SB, UK. [5]Biosciences Institute, Faculty of Medical Sciences, Newcastle University, Newcastle NE2 4HH, UK. [6]Department of Neurosurgery, Royal Victoria Infirmary, Newcastle upon Tyne NE1 4LP, UK. [7]Department for Clinical Neuroscience, NHS Lothian, Royal Infirmary Edinburgh, Edinburgh EH16 4SB, UK. [8]Translational Neurosurgery, Centre for Clinical Brain Sciences, University of Edinburgh, Edinburgh EH16 4SB, UK. ✉e-mail: sbooker@ed.ac.uk

labour and expansion of cell types, not observed in rodent neocortex[9]. Beyond PCs, the human neocortex contains a diverse population of inhibitory interneurons (INs) which make up to 21–44% of neurons in humans[10–12], which is higher than numbers observed in rodents[13], although most cell-types are broadly conserved[14]. Local INs form dense connections with PCs and each other to produce strong inhibition of their target cellular compartments via monosynaptic activation of GABA$_A$ receptors (GABA$_A$Rs[15]), and metabotropic GABA$_B$Rs[16,17].

GABA binds with high affinity to GABA$_B$Rs on principal cells following heterosynaptic spill-over from nearby inhibitory synapses[18–20], which activates inwardly rectifying K$^+$ (Kir3) channels to hyperpolarise neuronal membranes[21]. This K$^+$-conductance hyperpolarises dendrites over a long temporal window, but also displays prominent attenuation in electrotonically large neurons[22–24]. Somatodendritic GABA$_B$R activation directly reduces spike output[25], dendritic integration[26,27], and network oscillations[28,29]. In presynaptic compartments, GABA$_B$Rs inhibit voltage-gated Ca$^{2+}$-channels (VGCCs[21]) to regulate neurotransmitter release[16,30]. However, GABA$_B$Rs are not uniformly expressed in rodent brains, with a gradient of expression in both neocortex and hippocampus[22,31,32], a pattern which is not observed in postmortem human brain tissue[33,34], despite the presence of functional GABA$_B$Rs[3,35]. A major confound to human studies is whether tissue was collected from seizure-free individuals, as complex GABA$_B$R expression patterns are observed in different epilepsy patient groups[2–4]. Regardless, the role of GABA$_B$Rs in controlling cortical neuron and circuit function has not been fully characterised across the adult lifespan. We hypothesise that GABA$_B$R signalling in the human neocortex may display similarity to that of rodents in an age-dependent manner and be affected by a clinical history of seizures and/or anti-seizure medications.

To address this, we directly quantified functional GABA$_B$R signalling in the neocortex from rats and humans using whole-cell patch-clamp and extracellular local field potential (LFP) recordings. We show that GABA$_B$R signalling differs between L2/3 and L5 PCs in juvenile rats; which normalises into adulthood. Adult human cortical neurons display similar GABA$_B$R current densities to rodents, but possess stronger presynaptic inhibition, which combines to differentially control neuronal activity in the cortical column. At the circuit level, GABA$_B$Rs control the strength and synchrony of neuronal oscillations in human cortex. Finally, we show that GABA$_B$R signalling is enhanced in patients who receive the anti-seizure medication Levetiracetam (LEV)−irrespective of whether they experienced seizures prior to surgery. These findings highlight important phylogenetic differences in inhibitory signalling between rodents and humans, with direct implication for targeting GABA$_B$Rs in the treatment of seizure disorders.

## Results

### Divergent baseline excitability of PCs in seizure-free humans and rat neocortex

Human neurons are known to possess some divergence in electrophysiological properties from rodents[36,37]. To confirm this, we first compared the basal physiology of L2/3 and L5 neurons from primary somatosensory (S1) cortex of 1-, 6–8-, and 12–14-month-old rats, and neocortex of adult humans (Fig. 1A and B). Human data was obtained from recordings from 53 individuals who were aged between 28 and 77 years old (median: 57.5 years). All human brain tissue was obtained as 'access tissue' from individuals undergoing tumour resection. As seizure or anti-seizure medication history may influence neuronal excitability and neurotransmitter signalling, we first analysed data from only those patients that were seizure-free and had not received the anti-seizure medication LEV ($N = 29$ cases; Supplementary Table 1).

L2/3 neurons typically displayed fewer action potentials (APs) than L5 counterparts in response to depolarising current steps, irrespective of rat or human origin (Fig. 1B). However, when comparing the response of neurons to current inputs (FI slope), L2/3 PCs displayed increased excitability in rats as a function of age, which was not observed in L5 PCs (Fig. 1C). However when comparing FI slope between adult rats (6–14 month-old) and adult humans, we found no species difference in overall spike output, despite layer-specific effects (Fig. 1C'). In rats, membrane capacitance of L2/3 did show differences over the lifespan, with L2/3 possessing higher capacitance than L5 in 1-month-old rats ($t = 3.14$, $p = 0.003$, Tukey post hoc test), a relationship which was reversed in 6–8 months ($t = 3.6$, $p = 0.0006$, Tukey post hoc test) and 12–14-month-old rats ($t = 2.15$, $p = 0.036$, Tukey post hoc test, Fig. 1D). The pattern of L5 PCs displaying higher capacitance than L2/3 PCs was consistent when comparing between all adult rats and humans (Fig. 1D'). A summary of rat intrinsic physiology is shown in Supplementary Table 2 and a comparison of adult rat and human physiology is in Supplementary Table 3. As many of the key electrophysiological properties of neurons show age-dependent changes, we asked if such changes are present during adulthood in humans. To address this, we compared the age of patients to the electrophysiological properties of individual neurons. We found no significant correlation of any tested neuronal parameter with age (Supplementary Fig. 1).

### Divergent functional GABA$_B$R-mediated currents in human and rat neocortex

The human cortex is known to express functional GABA$_B$Rs[35] that may show age-dependent changes in expression at the total protein level[38]. Whether the function of these receptors deviates from those of rodents remains largely unexplored. To determine the functional expression of GABA$_B$Rs in neurons, we performed whole-cell recordings from identified L2/3 and L5 PCs from rats and humans in the presence of blockers of AMPA, NMDA, and GABA$_A$ receptors leaving GABA$_B$R signalling intact. Following baseline recording we applied the selective GABA$_B$R agonist baclofen (10 μM) to the bath to activate all receptors on recorded neurons, followed by the potent and selective GABA$_B$R antagonist CGP-55,845 (CGP, 5 μM, 5 min).

We observed a large outward current in neurons following 10 μM baclofen application, which was blocked by subsequent application of CGP (Fig. 2A). As expected from previous reports[24,26], we observed 64% lower whole-cell currents in L5 PCs in 1-month-old rats when quantified as the peak baclofen current ($t = 4.96$, $p < 0.0001$, Tukey post hoc test; Fig. 2B). In 6–8-month-old rats, L5 neurons had total currents largely similar to L2/3 ($t = 1.01$, $p = 0.317$, Tukey post hoc test). In 12-14-month-old rats, L5 neurons had total currents largely similar to L2/3 ($t = 1.68$, $p = 0.105$, Tukey post hoc test). As baclofen currents did not differ between adult rats from 6-14 months (L2/3: $t = 0.55$, $p = 0.85$, Tukey post hoc test; L5: $t = 1.64$, $p = 0.24$, Tukey post hoc test), we pooled these together into a single adult cohort. In human neurons, we observed minimal difference between L2/3 and L5 neurons which did not differ from adult rats (Fig. 2B'). As cell capacitance (a proxy for electrotonic size) differed between cell-types across rat age (Fig. 1E), we next normalised baclofen whole-cell currents to measure capacitance for each cell type across the rat lifespan (Fig. 2C). As for absolute currents, we found in 1-month-old rats had a baclofen current density was 63% lower in L5 PCs compared to L2/3 ($t = 4.19$, $p = 0.0002$, Tukey post hoc test). In adult rats, L5 PCs had a baclofen current density similar to that of L2/3 PCs at both 6–8 months ($t = 1.09$, $p = 0.28$, Tukey post hoc test) and at 12–14-month-old rats ($t = 0.31$, $p = 0.76$, Tukey post hoc test). There was no apparent difference in baclofen current density between adult rats and adult humans (Fig. 2C'). To test whether GABA$_B$R functional currents are impacted by age in rat and human neurons, as has been suggested in proteomic studies[38], we compared the effect of age on GABA$_B$R current density over human adulthood. Comparing the specific age of rats, we found a decrease in baclofen current density from early adolescence into adulthood in rats (Fig. 2D), but no age-dependent decline in humans between the ages of 28 and 77 for either L2/3 PCs or L5 PCs (Fig. 2E).

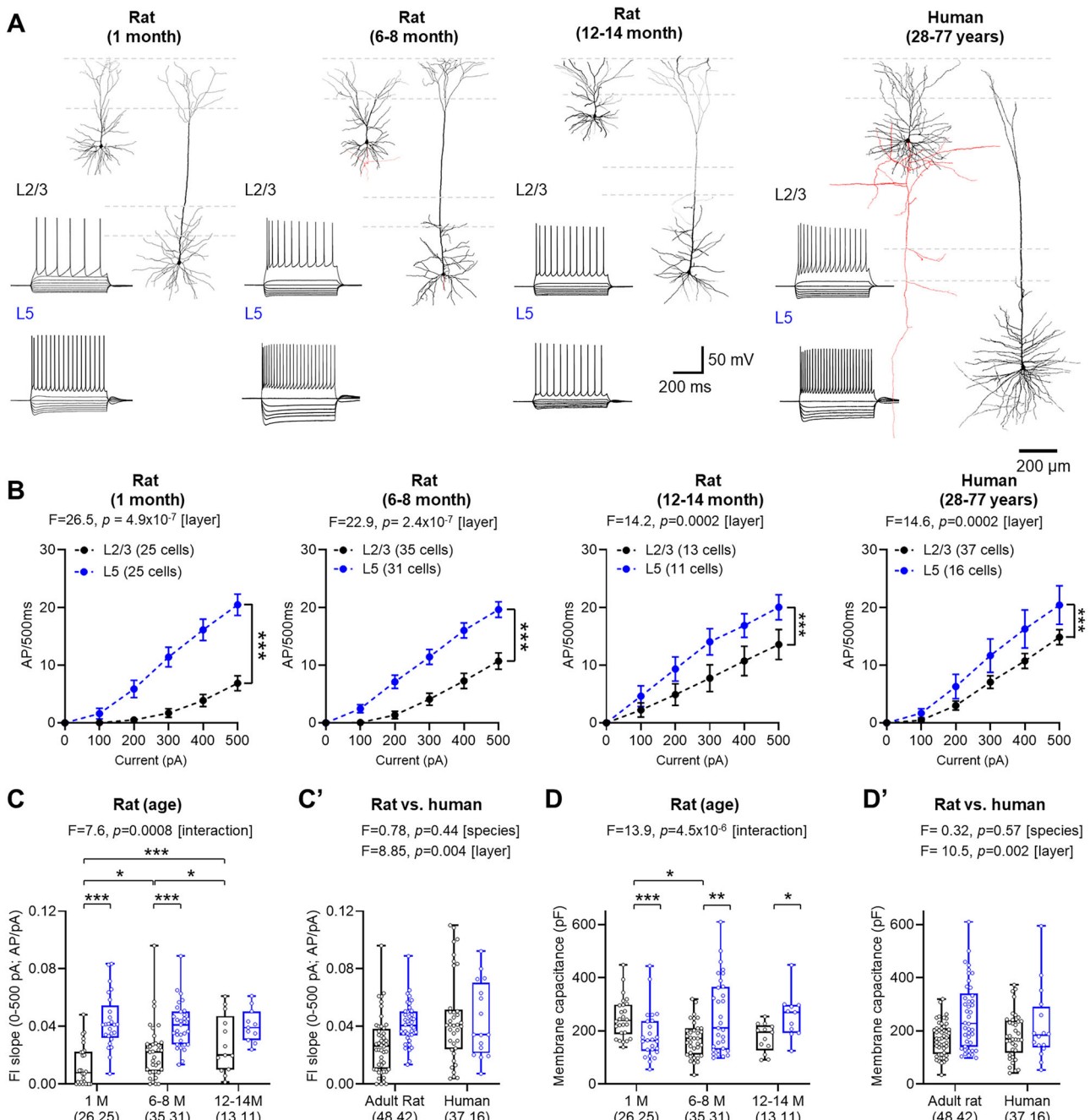

**Fig. 1 | Age- and species-dependent changes in neuronal excitability. A** Example reconstructions of L2/3 and L5 neurons from rat and human neocortex. Inset, voltage response of example cells in response to −500 to +500 pA current injections (100 pA steps, 500 ms duration). **B** Current-voltage responses of L2/3 (black) and L5 (blue) neurons from 1-month-old (left; L2/3: 25 cells from 14 rats, L5: 25 cells from 13 rats), 6–8-month-old (middle left; L2/3: 35 cells from 20 rats, L5: 31 cells from 20 rats), and 12–14-month-old rats (middle right; L2/3: 13 cells from 5 rats, L5: 11 cells from 5 rats), and human neocortex (right; L2/3: 37 cells from 20 cases, L5: 16 cells from 9 cases). **C** The slope of the current-frequency (FI) plot between 0 and 500 pA from recorded neurons between layers and age of the same rats (L2/3 vs. L5: $p = 2.6 \times 10^{-9}$ (1-month), $2.55 \times 10^{-9}$ (6–8 months), 0.599 (12–14 months), L2/3: $P = 0.0004$ (1 vs. 12–14 months), 0.0335 (1 vs. 6–8 months), 0.0191 (6–8 vs.

12–14 months); Tukey post hoc tests). **C'** Comparison of FI slopes of L2/3 and L5 PCs in adult rats (6–14 months) and humans. **D** Membrane capacitance of recorded neurons in rats for the same age groups (L2/3 vs. L5: $p = 0.0025$ (1-month), 0.0006 (6–8-month), 0.0364 (12–14 months), L2/3: $P = 0.0729$ (1 vs. 12–14 months), 0.0029 (1 vs. 6–8 months), 0.834 (6–8 vs. 12–14 months); Tukey post hoc tests). **D'** Comparison of membrane capacitance L2/3 and L5 PCs in adult rats (6–14 months) and humans. All data is shown as either mean ± SEM (**B**) or boxplots (25–75% range) with the median indicated and whiskers indicating maximum/minimum ranges (**C** and **D**). All data are overlain by responses of individual cells (circles). Statistical test results are shown above graphs from 2-way ANOVA (2-sided, **B**) and LMM models (**C** and **D**), with results from post-hoc 2-sided Tukey tests: *$P < 0.05$, **$P < 0.01$; ***$P < 0.001$.

In some recordings, cells displayed negative whole-cell currents relative to baseline when CGP was bath applied, indicative of a tonic GABA$_B$R current. To directly test this, in a subset of recordings we applied CGP alone to rat slices, without prior baclofen application (Fig. 2F). Consistent with baclofen whole-cell currents, we observed larger tonic currents in L2/3 compared to L5 PCs in 1-month-old rats, which was absent in 6–14 month-old rats (Fig. 2G), supporting the notion that layer-wise differences in GABA$_B$R signalling are normalised by adulthood. These data reveal that functional GABA$_B$Rs undergo an early life down-regulation in rats, with the greatest currents observed

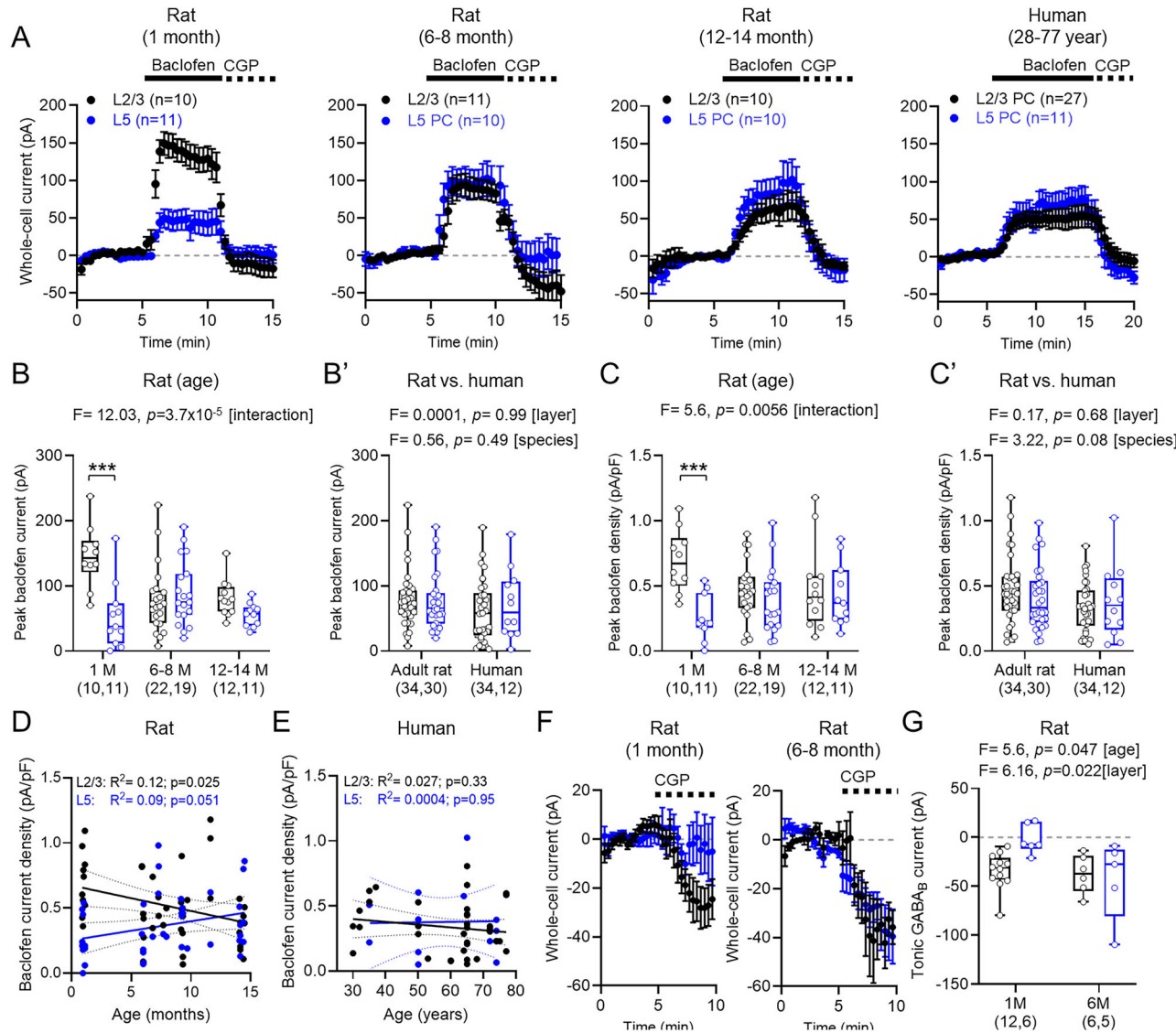

**Fig. 2 | Pharmacologically activated postsynaptic GABA$_B$R currents in rat and human neocortex. A** Whole-cell currents from L2/3 (black) and L5 (blue) PCs from 1-, 6–8-, 12–14-month-old rats, and human neocortex following bath-application of baclofen and CGP. Number of cells and baseline current (grey line) indicated. **B** Peak baclofen current from L2/3 and L5 in 1-month (L2/3: 10 cells/8 rats; L5: 11 cells/5 rats), 6–8-month (L2/3: 22 cells/13 rats; L5: 19 cells/13 rats), and 12–14-month-old (L2/3: 12 cells/5 rats; L5: 11 cells/5 rats) rats (L2/3 vs. L5: $p = 7.0 \times 10^{-6}$ (1-month), 0.317 (6–8-month), 0.105 (12–14-month). **B'** Peak baclofen current of adult rats and humans (L2/3: 34 cells/19 cases; L5: 12 cells/8 cases). **C** Baclofen current-density for rat neurons (L2/3 vs. L5: $p = 0.0002$ (1-month), 0.282 (6–8-month), 0.762 (12–14-month). **C'** Baclofen current density for adult rat and human neurons. **D** Scatter-plot of baclofen current density for rat L2/3 (black) and L5 (blue) and age

(months) showing linear regression (solid lines) with 95% confidence intervals (dashed lines) and $R^2$ and $P$-values ($F$-test). **E** Scatter-plot of baclofen current density from human L2/3 (black) and L5 (blue) plotted against age (years) in the same format. **F** Time-course plots of L2/3 (black) and L5 (blue) PCs from 1-month (L2/3: 8 cells/5 rats; L5: 6 cells/3 rats) and 6–8-month (L2/3: 8 cells/5 rats; L5: 7 cells/5 rats) following bath-application of CGP. **G** GABA$_B$R-mediated tonic-currents plotted for rat PCs. Data is shown as mean ± SEM (**A, F**), or boxplots (25–75% range) with median indicated and whiskers indicating maximum/minimum ranges (**B, C, G**) or individual data (filled circle) as a scatter-plot (**D, E**). Box-plots are overlain by responses of individual cells (open circles). Statistics shown from LMM (**B, C, G**) or Pearson correlation (**D, E**), with comparisons from 2-sided Tukey post-hoc tests: *$P < 0.05$, **$P < 0.01$; ***$P < 0.001$. Source data are provided as a Source Data file.

in L2/3 of juvenile rats. In seizure-free adult humans GABA$_B$R-mediated currents were similar between L2/3 and L5 PCs at all ages examined.

It has been shown previously that GABA$_B$Rs localise to the neocortex of rodents[31] and humans[33]. To qualitatively confirm the distribution of GABA$_B$Rs in cortical columns, we performed immunohistochemical labelling for GABA$_{B1}$ subunits in fixed samples of 1, 6–8, and 12-14-month-old rats and humans (Fig. 3A). Measurement of the fluorescence intensity of GABA$_{B1}$ labelling (z-scored) was plotted against the normalised cortical column, to account for differences in thickness. This revealed in rodents (Fig. 3B), that GABA$_{B1}$ labelling was highest in L1 and diminished in intensity towards the white matter. By

contrast, in humans we observed high levels of labelling close to the pia and L1, which transiently reduced, then increased again towards the deeper cortical layers (Fig. 3C). Notably, in human cortex we observed a high degree of somatic labelling for GABA$_{B1}$, which was not observed in rodents. These data suggest that while GABA$_B$Rs are expressed throughout the cortical column, subtle species-specific differences may exist.

We next determined whether GABA$_B$R-mediated currents in L2/3 and L5 neurons were driven by endogenous GABA release. To achieve this, we delivered short trains of high-frequency stimulation to the L1/2 border (5 stimuli, 200 Hz, 50 V) in the presence of the same ionotropic

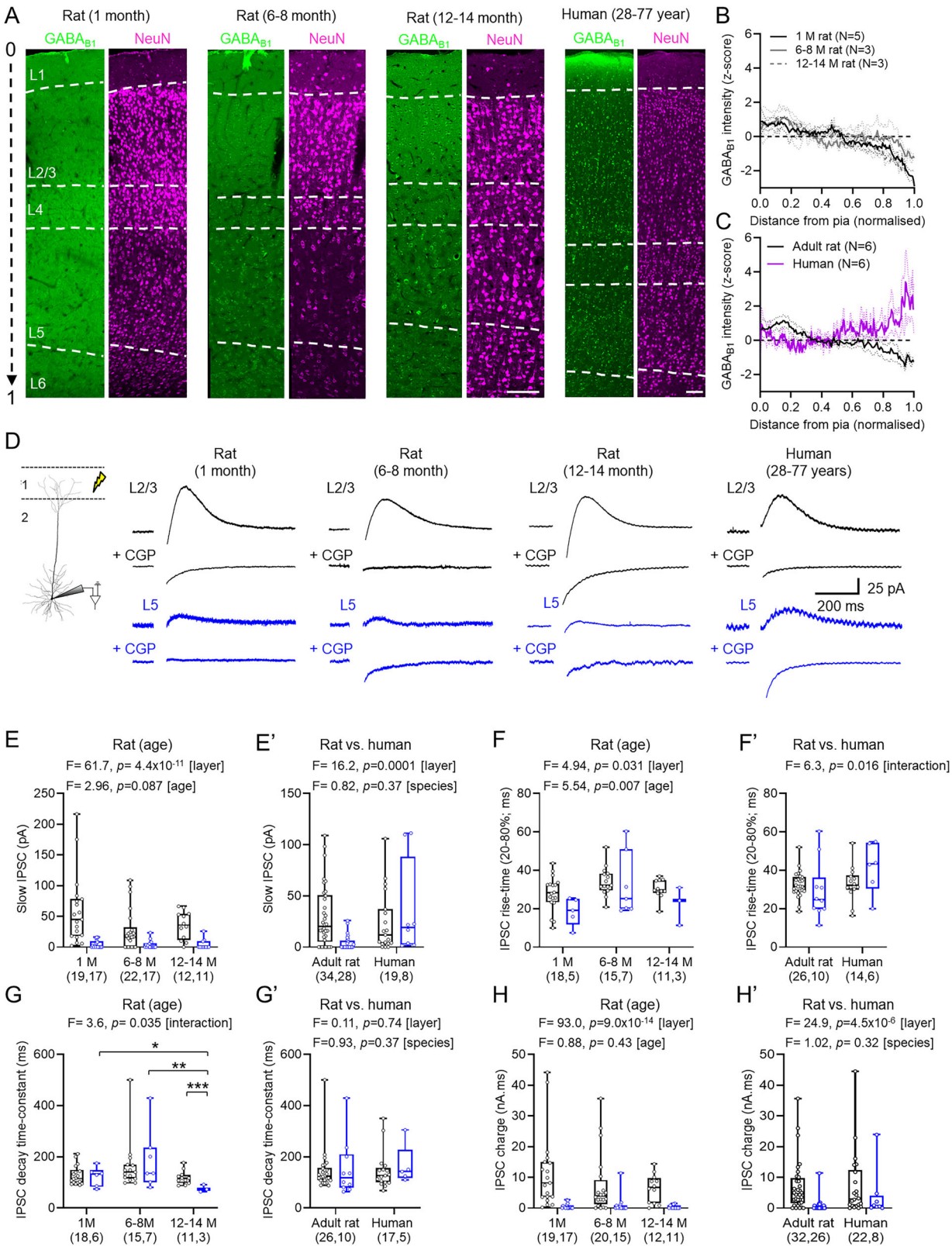

receptor blockers as above. In L2/3 neurons from all species and ages, electrical stimulation elicited large amplitude, slow IPSCs. By comparison, in L5 PCs we largely observed small amplitude slow IPSCs following L1 stimulation. All slow IPSCs were sensitive to application of CGP (Fig. 3D). IPSCs elicited in 1-month-old rats typically had large GABA$_B$R-mediated slow-IPSCs in L2/3 PCs compared to L5 PCs, which were also observed at 6–8 and 12–14-months (Fig. 3E). A similar pattern

was observed in human L2/3 and L5 neurons (Fig. 3E') The combination of GABA$_B$Rs, effector channels, and auxiliary proteins can alter receptor kinetics[39], as well as electrotonic properties[22], thus we next measured slow-IPSC kinetics. The 20–80% rise time of IPSCs was consistently faster in L5 PC than L2/3 PCs in rats, which was modulated by age (Fig. 3F). The rise time of GABA$_B$R-mediated IPSCs displayed an interaction of species and layer when comparing human to adult rat,

**Fig. 3 | GABA release activates GABA$_B$Rs with age-dependent differences.**
**A** Immunofluorescence for GABA$_{B1}$ (green) and NeuN (magenta) in rat and human neocortex, with layers (dashed lines) and depth (arrow). Scale = 100 μm.
**B** GABA$_{B1}$ intensity normalised to cortical depth for 1 (black, $n = 5$), 6–8 (grey, $n = 3$) and 12–14-month-old (grey, $n = 3$) rats, with SEM (dotted lines). **C** GABA$_{B1}$ intensity for human (magenta, 6 patients) and adult rat (black, 6–14-months).
**D** Schematic of IPSC recordings, with control and CGP from L2/3 (black) and L5 (blue). **E** IPSC amplitudes in 1 (L2/3: 19 cells/11 rats; L5: 17 cells/8 rats), 6–8 (L2/3: 22 cells/13 rats; L5: 17 cells/13 rats), and 12–14-month-old (L2/3: 12 cells/5 rats; L5: 11 cells/5 rats) rats. **E'** IPSC amplitude for adult rats and humans (L2/3: 19 cells/13 cases; L5: 8 cells/5 cases). **F** IPSC 20–80% rise-time in 1-month (L2/3: 18 cells/10 rats; L5: 5 cells/5 rats), 6–8-month (L2/3: 15 cells/13 rats; L5: 7 cells/6 rats), and 12–14-month-old (L2/3: 11 cells/5 rats; L5: 3 cells/2 rats) rats. Rise-time could not be measured for all IPSCs. **F'** 20–80% rise-time in adult rat and humans (L2/3: 14 cells/10 cases; L5: 6 cells/4 cases). **G** IPSC decay time-constant from 1 (L2/3: 18 cells/10 rats; L5: 6 cells/5 rats), 6–8 (L2/3: 15 cells/13 rats; L5: 7 cells/6 rats), and 12–14 (L2/3: 11 cells/5 rats; L5: 3 cells/2 rats) rats (L5: 1 vs. 12–14 months, $p = 0.035$, L5: 6–8 vs.12–14 months, $p = 0.0013$; 12–14-month: L2/3 vs. L5: $P = 0.0048$). **G'** Decay time-constants of adult rat and human (L2/3: 17 cells/10 cases; L5: 5 cells/5 cases). **H** IPSC charge-transfer 1 (L2/3: 19 cells/11 rats; L5: 17 cells/8 rats), 6–8 (L2/3: 20 cells/13 rats; L5: 15 cells/13 rats), and 12–14-month-old (L2/3: 12 cells/5 rats; L5: 11 cells/5 rats) rats. **H'** IPSC charge-transfer for adult rats and humans (L2/3: 22 cells/13 cases; L5: 8 cells/5 cases). Data shown as boxplots (25–75% range) with median indicated and whiskers indicating maxima/minima, showing individual points. Statistics from LMM with 2-sided Tukey post-hoc tests: *$P < 0.05$, **$P < 0.01$; ***$P < 0.001$. Source data are provided as a Source Data file.

which appeared due to divergences in L5 PCs, but was not significantly different ($t = 2.15$, $p = 0.091$, Tukey post hoc test, Fig. 3F'). The decay time-constant of evoked slow-IPSCs was faster in L5 PCs from 12–14-month-old rats than L2/3 PCs ($t = 2.95$, $p = 0.005$, Tukey post hoc test) or in 1-month ($t = 2.57$, $p = 0.035$, Tukey post hoc test) or 6–8-month ($t = 3.76$, $p = 0.001$, Tukey post hoc test) rats (Fig. 3G). Overall, when comparing adult rats to humans, we found no layer or species difference in GABA$_B$R IPSC decay time-constants (Fig. 3G'). The subtle effects on kinetics could lead to differences in the total charge of GABA$_B$R-mediated IPSCs. We found no effect of rat age (Fig. 3H) or species (Fig. 3H') on the charge transfer of slow IPSCs.

These data show that GABA$_B$Rs are reliably recruited by GABA release in human and rat cortical circuits, giving rise to slow-IPSCs which are largely consistent between species, showing no differences in amplitude and kinetics.

## GABA$_B$Rs display developmentally divergent contributions to perisomatic inhibition

GABA$_B$Rs in juvenile cortical neurons have been shown to largely localise to distal dendritic compartments, and thus may have minimal direct influence on perisomatic excitability[24,26]. Given age differences in postsynaptic GABA$_B$R-mediated currents, we asked if these receptors differentially modulate somatic excitability. For this we focally applied baclofen (50 μM, 200 ms) directly to the perisomatic region of neurons (Fig. 4A). Perisomatic baclofen puff application gave rise to outward currents in recorded L2/3 and L5 cells in rat and human neurons (Fig. 4B), which were blocked by bath application of CGP (Fig. 4D, F, and H). To determine whether perisomatic GABA$_B$Rs were able to control neuronal activity we puff applied baclofen to neurons entrained to tonic firing (rheobase + 25 pA). Focal baclofen application to the perisomatic region consistently reduced the AP discharge of L2/3 cells, independent of age or species. The observed reduction in AP discharge was blocked by CGP application (Fig. 4C, E, and G). The same focal GABA$_B$R activation in L5 pyramidal cells inhibited discharge in 1-month-old rats, less so than for L2/3 PCs ($p = 0.002$, Mann–Whitney test; Fig. 4D). In 6–8-month-old rats, the same baclofen puff produced near complete loss of firing in L5 PCs, comparable to L2/3 PCs ($p = 0.65$, Mann–Whitney test; Fig. 4F); which was also observed in humans ($p = 0.87$, Mann–Whitney test; Fig. 4H).

These data confirm that GABA$_B$Rs located in the perisomatic compartment more efficiently control the output of L5 neurons in adult human and rat cortex. These age-dependent effects are likely due to increased perisomatic localisation of the receptor and the hyperpolarisation produced by its activation in L5 PCs over later life.

## GABA$_B$Rs more strongly inhibit presynaptic glutamate release in human cortex, compared to rodents

At glutamatergic synapses, presynaptic GABA$_B$Rs rely on heterosynaptic spillover from neighbouring GABAergic afferents, thus providing powerful circuit-wide control of neurotransmitter release[16,20]. We next quantified the magnitude of GABA$_B$R presynaptic inhibition at major glutamatergic inputs to L2/3 (Fig. 5A) and L5 (Fig. 5E) PCs in adult rat and human cortex. To achieve this, we performed whole-cell patch clamp recordings from L2/3 and L5 PCs at −70 mV voltage-clamp in the presence of picrotoxin (50 μM). We evoked excitatory postsynaptic synaptic currents (EPSCs) in neurons by stimulating L1 (Fig. 5B) or L4 (Fig. 5C) for afferents to L2/3 PCs, or L1 (Fig. 5F) and L5 (Fig. 5G) for afferents to L5 PCs. To assess the effect of GABA$_B$Rs on evoked EPSCs, 10 μM baclofen was applied to the bath for 5 min, followed by 5 μM CGP-55,845. These experiments were also performed in 6–8-month (Supplementary Fig. 2) and 1-month-old rats (Supplementary Fig. 3). All experiments used pairs of stimuli (50 ms interval) to assess the paired-pulse ratio (PPR; Supplementary Fig. 4).

In human L2/3 neurons, stimulation of L1 and L4 produced EPSCs which displayed consistent facilitating paired-pulse ratios (PPR, Fig. 5B, C, F, and G). Bath application of baclofen resulted in a very strong, near complete loss of EPSCs at both L1 and L4 inputs, which was fully recovered by CGP application. Similarly, in L5 PCs we observed a near-complete inhibition of L1 and L5 inputs following the baclofen wash-in, which was recovered by CGP application. When we performed the same recordings in adult (6-8-month, Supplementary Fig. 2) rats we also observed reductions in EPSC amplitude following baclofen application, but lower than in human L2/3 PCs (Fig. 5D). Likewise, we found that presynaptic GABA$_B$R mediated inhibition in human L5 afferents was stronger than for rats, irrespective of afferent type (Fig. 5H). Presynaptic GABA$_B$Rs also similarly blocked afferents to L2/3 and L5 PCs in 1-month-old rats (Supplementary Fig. 3). To confirm that endogenous GABA release was capable of activating presynaptic GABA$_B$Rs in human cortex, we performed a subset of experiments in which trains of 10 EPSCs were elicited in L2/3 PCs following L1 stimulation at either 10 or 20 Hz (Fig. 5I). 10 stimuli resulted in weakly depressing EPSCs at both 10 Hz (Fig. 5J) and 20 Hz (Fig. 5K), the depression of which was reduced by bath application of CGP. These data confirm that endogenous GABA release is sufficient to recruit presynaptic GABA$_B$Rs in human cortex.

With respect to GABA$_B$R activation, most synaptic inputs to L2/3 and L5 PCs displayed increased PPR, and thus reduced release probability, upon baclofen application (Supplementary Fig. 4). Such PPR changes were most prominent at human synapses, especially onto L2/3 PCs—consistent with enhanced presynaptic inhibition. Subsequent CGP applications typically block baclofen-dependent PPR increases. Taken together, these data show that GABA$_B$R presynaptic inhibition is stronger in presynaptic afferents terminating onto human cortical neurons. Given the greater overall inhibitory tone, this suggests that presynaptic GABA$_B$Rs may strongly inhibit active human cortical circuits.

## Population oscillations in human cortex are highly sensitive to GABA$_B$R activation

So far, we have shown that GABA$_B$Rs can strongly inhibit both pre- and postsynaptic domains of L2/3 and L5 PCs of the adult human cortex. As endogenous GABA$_B$R activation requires heterosynaptic spill-over of GABA[19], the effects of GABA$_B$R activation may be most prominent in

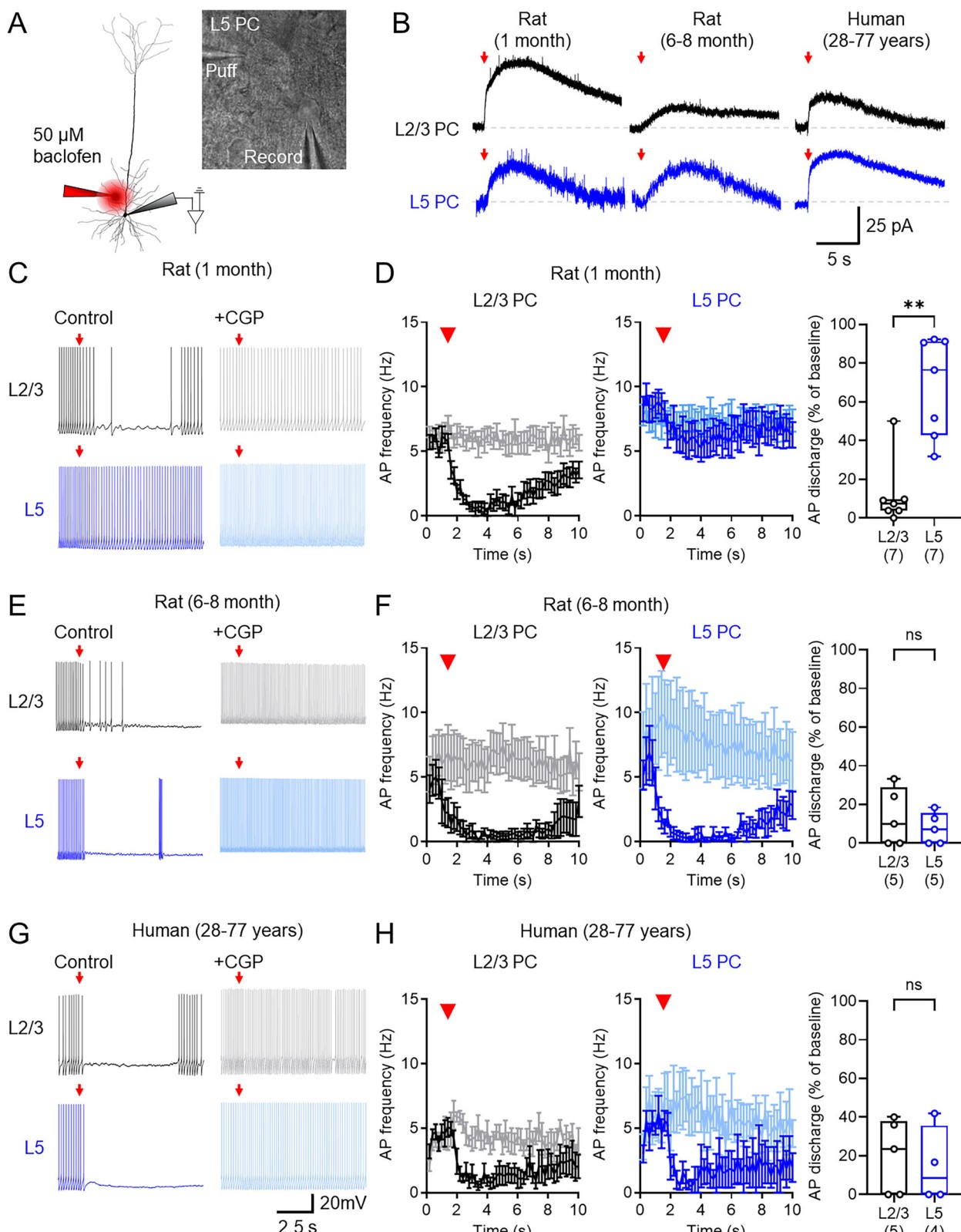

active circuits[20]. To determine the circuit-level effects of GABA_BR activation, we conducted local field potential (LFP) recordings to assess how in vitro neuronal oscillations are affected by baclofen. Persistent neuronal oscillations were pharmacologically induced using kainic acid (KA, 0.6–2 μM) and carbachol (CCh, 60–200 μM), as previously described in rodent[40–42] and human cortical tissue[43]. Following 30–40 min of KA/CCh application we reliably observed broadband

neuronal oscillations in L2/3 and L5 of human neocortex (Fig. 6A), with Fast-Fourier transform displaying prominent oscillations in theta-alpha (4–10 Hz), beta (10–30 Hz), low-gamma (30–50 Hz), and high-gamma (51–85 Hz) frequency bands, which were strongly reduced by baclofen in a dose-dependent manner (Fig. 6B). Importantly, bath application of the GABA_BR antagonist CGP-55,845 (5 μM) did not reduce oscillatory power, and in fact increased the strength of

**Fig. 4 | Perisomatic GABA_BRs differentially inhibit AP discharge in L2/3 and L5 PCs over development. A** Schematic of experimental set-up depicting puff-application of 50 µM baclofen (red) to the perisomatic region during whole-cell recordings. IR-DIC image confirming somatic recording (record) and puff-pipette location in L5. **B** Baclofen puff-mediated currents recorded at −65 mV from L2/3 (black) and L5 PCs (blue), from 1- and 6–8-month-old rats, and humans. Baclofen puff (red arrow) and baseline (grey line) are shown for reference. **C** Action potential (AP) output of L2/3 (black, upper) and L5 (blue, lower) PCs from 1-month old rats held at +25 pA above rheobase following baclofen puff under control conditions (left) or in CGP (right). **D** AP firing during baclofen puff recordings for L2/3 (7 cells/4

rats) and L5 PCs (7 cells/4 rats), compared in the presence of CGP (grey or light blue). Right, AP discharge at 2–3 s after baclofen puff ($U = 2$, $p = 0.0023$). Number of cells is shown in parentheses. **E, F** The same analysis performed in 6–8-month-old rats (L2/3: 5 cells/4 rats; L5: 5 cells/4 rats; $U = 10$, $p = 0.651$). **G, H** The same analysis performed in human cortex (L2/3: 6 cells/5 cases; L5: 5 cells/5 cases; $U = 9$, $p = 0.873$). Data is shown as either mean ± SEM (left and middle panels) or boxplots (25−75% range) with median indicated and whiskers indicating maximum/minimum ranges with individual cell data overlain (right panels). Statistics are shown as ns $P > 0.05$, **$P < 0.01$, all from 2-sided Mann−Whitney tests. Source data are provided as a Source Data file.

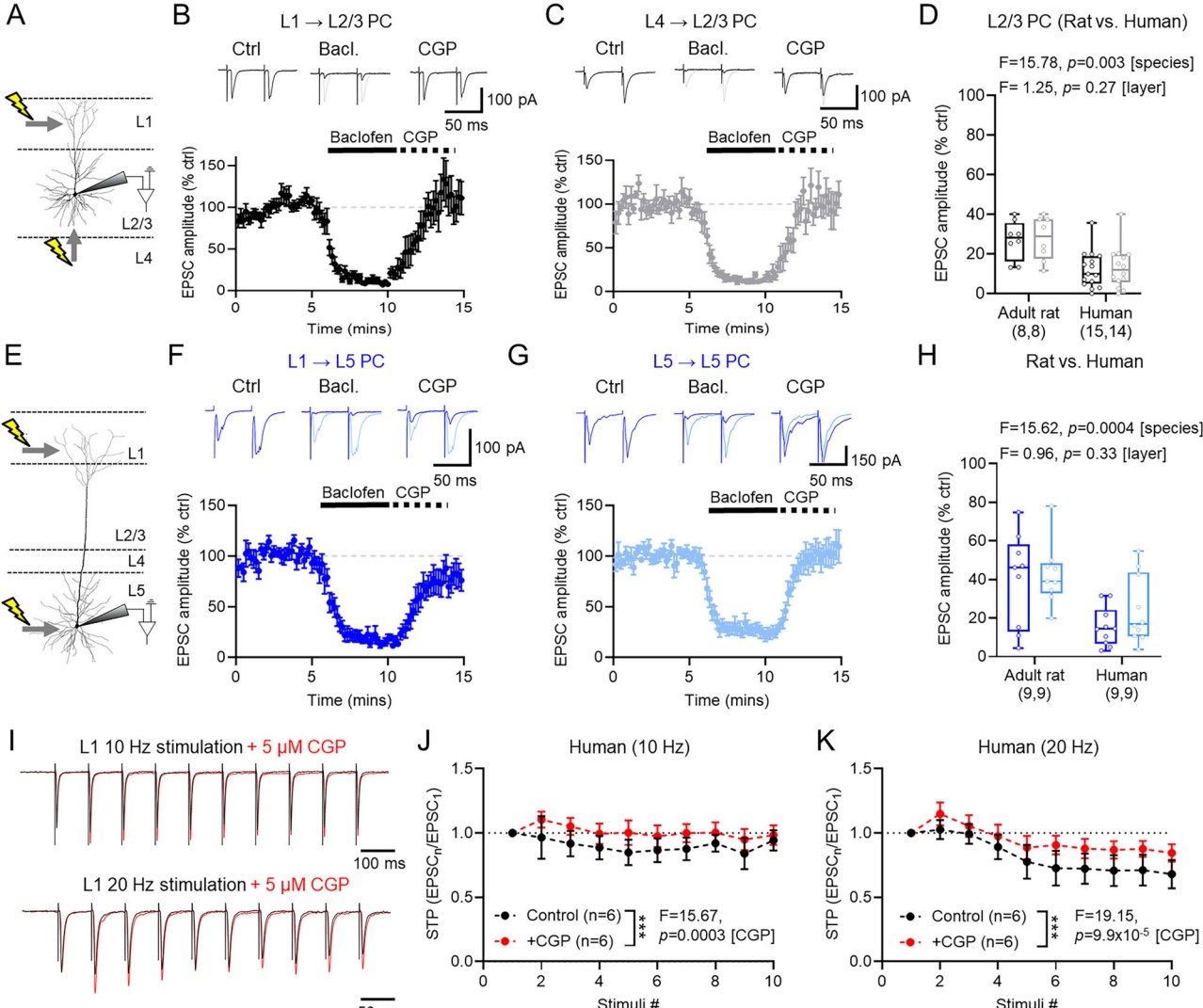

**Fig. 5 | Presynaptic GABA_BRs strongly inhibit synaptic inputs to human cortical PCs. A** Recording configuration for L2/3 PCs, showing stimulus (lightning) applied to L1 or L4. **B** EPSCs evoked by L1 stimulation in L2/3 PCs were recorded at −70 mV for control (Ctrl) and following application baclofen (Bacl.) and CGP, the latter underlain by control (grey). Time-course of EPSC amplitude following baclofen and CGP wash-in is shown below, with baseline indicated (grey dashed line). **C** Data in the same form, but for L4 inputs (light grey). **D** Baclofen-mediated inhibition of L1 (black) and L4 (grey) inputs to L2/3 PCs during the last minute of wash-in from 6–8-month-old rats (8 cells/4 rats) and humans (L1: 15 cells/7 cases; L4: 14 cells/7 cases). **E** Recording configuration for L5 PCs with stimulus applied to L1 or L5. **F** EPSCs evoked by L1 stimulation in L5 PCs under control (Ctrl), and following baclofen, and CGP, the latter underlain by control (blue traces). Time-course of EPSC amplitude following baclofen and CGP wash-in

is shown below. **G** Data in the same form, but for L5 inputs (light blue).
**H** Baclofen-mediated inhibition of L1 (blue) and L5 inputs (light blue) inputs to L5 PCs during the last minute of wash-in from 6–8-month-old rat (L1: 9 cells/4 rats; L5: 9 cells/4 rats) and human (L1: 10 cells/3 cases; L5: 10 cells/3 cases). **I** Human EPSC recordings from L1 stimulation in L2/3 PC following L1 stimulation, before (black) and after (red) CGP wash-in for 10 EPSCs have been amplitude scaled for brevity to clarify CGP effects. **J** Short-term plasticity (STP) of EPSCs, normalised to the first, at 10 Hz (6 cells/3 cases). **K** STP of EPSCs at 20 Hz (6 cells/3 cases). Data is shown as mean ± SEM (**B, C, F, G, J, K**) or boxplots (25–75% range) with the median indicated and whiskers indicating maximum/minimum ranges (**D, H**). All statistics are shown from either LMM (**D, H**) or 2-way repeated measures ANOVA (2-sided, **J, K**). Source data are provided as a Source Data file.

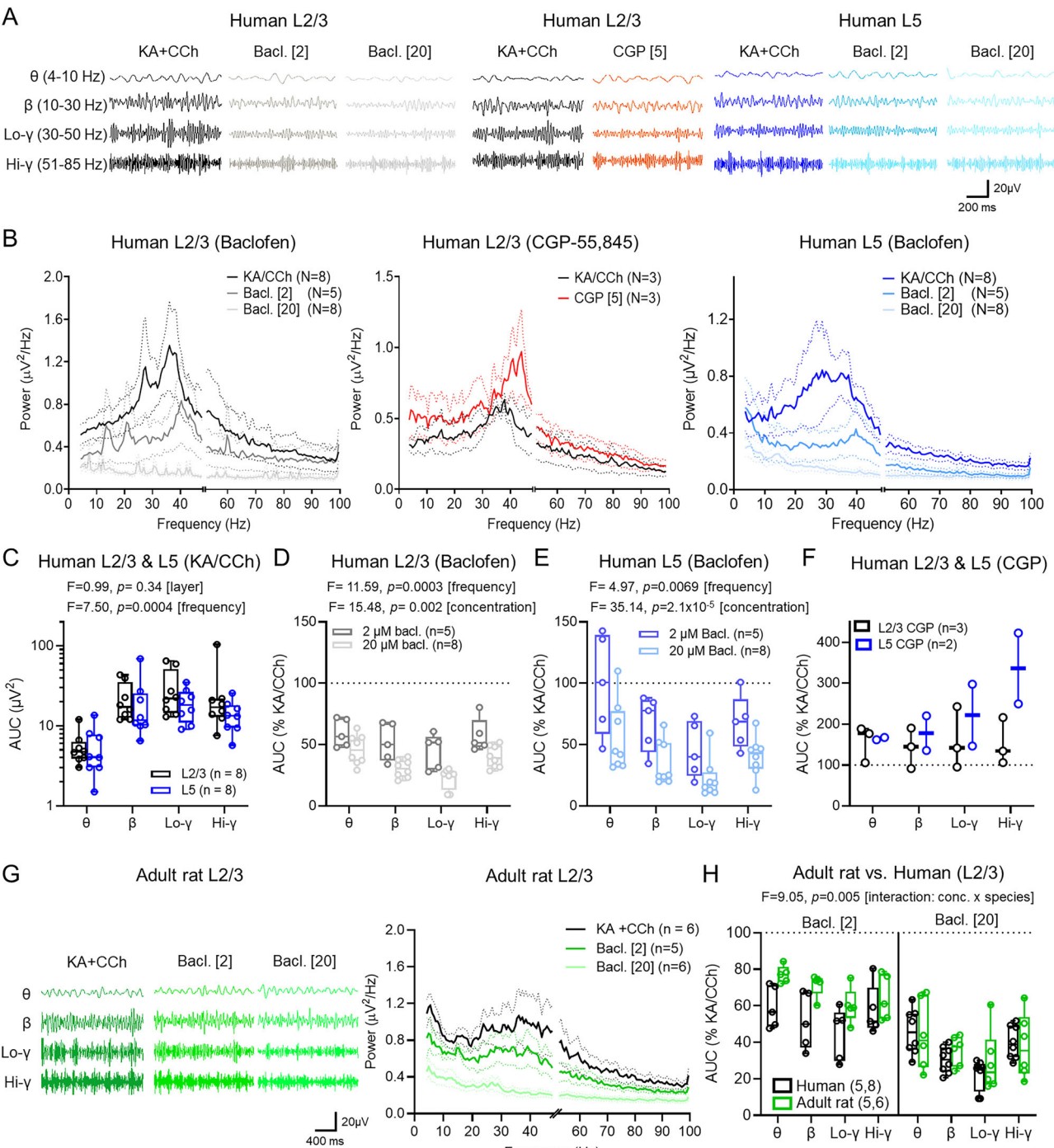

**Fig. 6 | GABA_BR activation attenuates oscillatory activity in L2/3 and L5 of the human cortex. A** Local field potential (LFP) signals filtered into theta ($\theta$), beta ($\beta$), low-gamma (Lo-$\gamma$), or high-gamma (Hi-$\gamma$) frequency ranges for L2/3 (black) and L5 (blue) during control (kainic acid and carbachol−KA+CCh) and during the last 2 min of 2 µM (light blue and grey) or 20 µM (lightest blue and grey) baclofen (Bacl.) application. **B** Power-spectra of L2/3 control (black; 8 slices/8 cases), 2 µM (grey; 5 slices/5 cases) or 20 µM (light grey; 8 slices/8 cases) baclofen recording, or following CGP (red; 3 slices/3 cases), or in L5 for control (blue; 8 slices/8 cases) 2 µM (light blue; 5 slices/5 cases) and 20 µM (lightest blue; 8 slices/8 cases) baclofen. **C** Area-under-curve (AUC) for KA+CCh recordings from L2/3 (black) and L5 (blue). **D** Effect on AUC (% change from KA+CCh) for frequency bands following 2 µM (dark grey, 5 slices/5 cases) and 20 µM (light grey; 8 slices/8 cases) baclofen. **E** The

same data but for L5 (2 µM Bacl.; light blue, 5 slices/5 cases) and (20 µM Bacl.; lightest blue; 8 slices/8 cases). **F** CGP wash-in on AUC change for L2/3 (3 slices/3 cases) and L5 (2 slices/2 cases), not tested due to insufficient replicates. **G** Filtered LFP signals from L2/3 of 6–8-month-old rats in KA+CCh (dark green), 2 µM (mid green) or 20 µM (lightest green) baclofen. Right, power spectra of adult rat LFP recording for control (black; 6 slices/6 rats) and 2 µM (mid green; 5 slices/5 rats), or 20 µM baclofen (mid green; 6 slices/6 rats). **H** Comparison of AUC change following 2 µM (left) and 20 µM (right) baclofen for L2/3 of human (black) and rat (green) recordings. Data are shown as mean ± SEM (**B**, **G**), or boxplots (25−75% range) with the median indicated and whiskers indicating maximum/minimum ranges with individual data points overlain. Statistics are shown from 2-way ANOVA (2-sided, **C**−**E**) or 3-way ANOVA (**H**). Source data are provided as a Source Data file.

oscillations in L2/3. Similar oscillatory activity and baclofen sensitivity was also observed in simultaneous recordings from L5 (Fig. 6B). We found no overt difference in the strength of KA + CCh-induced oscillations between L2/3 and L5 (Fig. 6C).

GABA$_B$R activation has been shown to abolish theta and gamma oscillations in rodent brain slices[28,29] and with complex actions on human EEG traces[44]. Following the generation of stable oscillations, we first applied baclofen at 2 µM, a low concentration likely to impact presynaptic receptors selectively[45]. In L2/3, baclofen led to a large reduction in oscillatory power over the main frequency bands tested 30–40 min after application. Subsequent bath application of 20 µM baclofen, a saturating concentration, produced further large reductions in oscillatory power at all frequencies (Fig. 6D). A similar pattern of inhibition was observed in LFP recordings from L5 (Fig. 6E). Antagonism of GABA$_B$Rs with CGP-55,845 (5 µM) tended to increase oscillatory power in a subset of L2/3 and L5 recordings (Fig. 6F).

As we found similar post-synaptic, but lower presynaptic, GABA$_B$R inhibition in rat PCs, we next asked whether baclofen differentially altered cortical oscillations in L2/3 of the adult rat. KA/CCh reliably produced similar oscillations in L2/3 of the adult rat, which were also sensitive to baclofen in a dose-dependent manner (Fig. 6G). Species-dependent comparison of the relative baclofen-mediated inhibition on cortical oscillations in L2/3 revealed that all oscillation frequencies were inhibited less by a low, presynaptic-selective[28] concentration of baclofen, when compared to humans, with similar inhibition observed at higher concentrations (Fig. 6H). Together, these data show that GABA$_B$R activation strongly controls the strength of oscillatory activity in the human cortex, with presynaptic GABA$_B$Rs differentially modulating cortical function in humans compared to rodents.

## GABA$_B$R activation reduces the synchrony of human cortical oscillations

Distinct neuronal oscillations interact with and synchronise each other, a process termed cross-frequency coupling[46], where neuronal firing patterns align to produce coordinated circuit-level activity. In the human neocortex, we found that low concentrations of baclofen strongly inhibited the oscillations across a range of frequencies, indicating the recruitment of presynaptic GABA$_B$Rs, which may impair direct neuronal signalling. We next asked if GABA$_B$R activation leads to reduced correlation of different oscillatory frequencies. For this, we performed more in-depth analysis of LFP recordings and how human cortical oscillations interact (Fig. 7A). First, autocorrelation analysis revealed that each of the oscillatory bands were highly consistent for both L2/3 and L5 (Fig. 7B). Next, we performed cross-correlation analysis between different frequency bands, notably theta vs. beta, theta vs. low-gamma, beta vs. low-gamma, and beta vs. low-gamma, under control conditions and following 2 and 20 µM baclofen in L2/3 (Fig. 7C). Similar cross-correlations were observed in L5 (Supplementary Fig. 3). In L2/3, we found that baseline correlation strength varied widely between frequency pairs, and was highly sensitive to baclofen application (Fig. 7D). For L5, we also observed high variability between frequency, which was highly sensitive to baclofen application (Fig. 7E). Similar baclofen sensitivity to cross-correlation strength was observed in L2/3 of the somatosensory cortex of adult rats (Supplementary Fig. 6). We found that there was minimal synchrony between L2/3 and L5 when cross-correlation was performed in a frequency-specific manner between layers, while some slices showed prominent correlation, particularly in the low-gamma range, they were not affected by baclofen application (Supplementary Fig. 7).

Finally, we examined phase-amplitude coupling (PAC) between these oscillation frequency pairs. PAC represents one form of cross-frequency correlation, wherein the phase of a lower frequency oscillation modulates the amplitude of a higher frequency[46]. PAC plays a crucial role in orchestrating coordinated local and global neuronal activity across spatial and temporal scales[47,48]; especially

theta/low-gamma and beta/low-gamma, which are associated with memory encoding[49] and sensorimotor function[50]. We found that theta vs. low-gamma and beta vs. high-gamma oscillatory pairs possessed prominent phase-amplitude coupling, but this was not affected by the application of baclofen (Supplementary Fig. 8).

These data indicate that GABA$_B$Rs contribute to the synchronous activity of local cortical circuits, including the interactions between oscillatory frequencies. Given the role of somatostatin and parvalbumin interneurons[51,52] in the generation of beta and gamma oscillations, it is plausible that such loss of inter-frequency correlation indicates the possibility of cell-type specific effects. As perturbations in PAC strength have been linked to cognitive impairments, such as in learning and memory[49] the loss of beta and low-gamma synchrony indicates that even modest concentrations of baclofen may impair cortical circuit function in vivo.

## Clinical presentation of seizures is correlated with higher GABA$_B$R activation

A key consideration of human resected brain tissue is seizure history[53], which may have an impact on GABA$_B$R expression[2–4,54]. So far, we have only presented data from patients who did not experience seizures in their recent clinical history, however, tissue was also collected from 15 patients who had experienced seizures and been prescribed LEV (1 g/day) at least 3 days prior to surgery. In addition, 4 patients had been prescribed 1 g/day LEV but had not experienced seizures. We interrogated our data to ask what effect a recent history of seizures and/or LEV treatment may have on neuronal activity and GABA$_B$R signalling. For this, we compared this data to that collected under identical circumstances from L2/3 neurons from seizure-free individuals (Supplementary Table 1).

Consistent with the administration of acute LEV[55], we observed no difference in overall AP discharge responses in L2/3 PCs from individuals who had received LEV (Fig. 8A and B). However, we observed that LEV treatment was associated with hyperpolarised membrane potentials, while other intrinsic physiological parameters were unaffected (Supplementary Table 3).

Assessing GABA$_B$R function, we first examined slow-IPSCs resulting from L1/2 stimulation (as in Fig. 3). We found no overall difference in GABA$_B$R-mediated IPSC amplitudes between any patient group (Fig. 8C). We found that wash-in of baclofen revealed greater whole-cell currents in patients experiencing seizures and receiving LEV than that of LEV-free individuals (Fig. 8D). Quantification of GABA$_B$R current density revealed that patients who had experienced seizures had 59% greater baclofen-mediated currents than control patients ($t = 2.96$, $p = 0.017$, Tukey $post$ $hoc$ test), and LEV/seizure-free patients tended to have greater baclofen current densities, albeit not significantly so ($t = 2.05$, $p = 0.129$, Tukey post hoc test; Fig. 8E). To determine whether acute LEV increases GABA$_B$R currents, we performed a separate set of experiments in which human brain slices were incubated with 100 µM LEV for 3 h prior to recording, a concentration similar to the expected CSF concentration in patients receiving 1 g/day[56,57]. We found that in brain slices from control patients and those who had received LEV prior to surgery, subsequent in vitro application of LEV did not lead to enhanced baclofen-mediated currents, despite overall patient group differences replicating our previous finding (Fig. 8F).

To ascertain whether experiencing seizures and administration of LEV altered circuit function, we assessed oscillatory activity following KA+CCh application. We found no overt difference in oscillatory power between patients, irrespective of clinical background (Fig. 8G). As patients who had not experienced seizures, but had been prescribed LEV, displayed the greatest GABA$_B$R-mediated signalling (compared to control), we asked how baclofen affected oscillations in brain slices from these patients. Comparison of the change in power, as measured as % difference from the control of the AUC, revealed that patients who had experienced seizures and received LEV did not

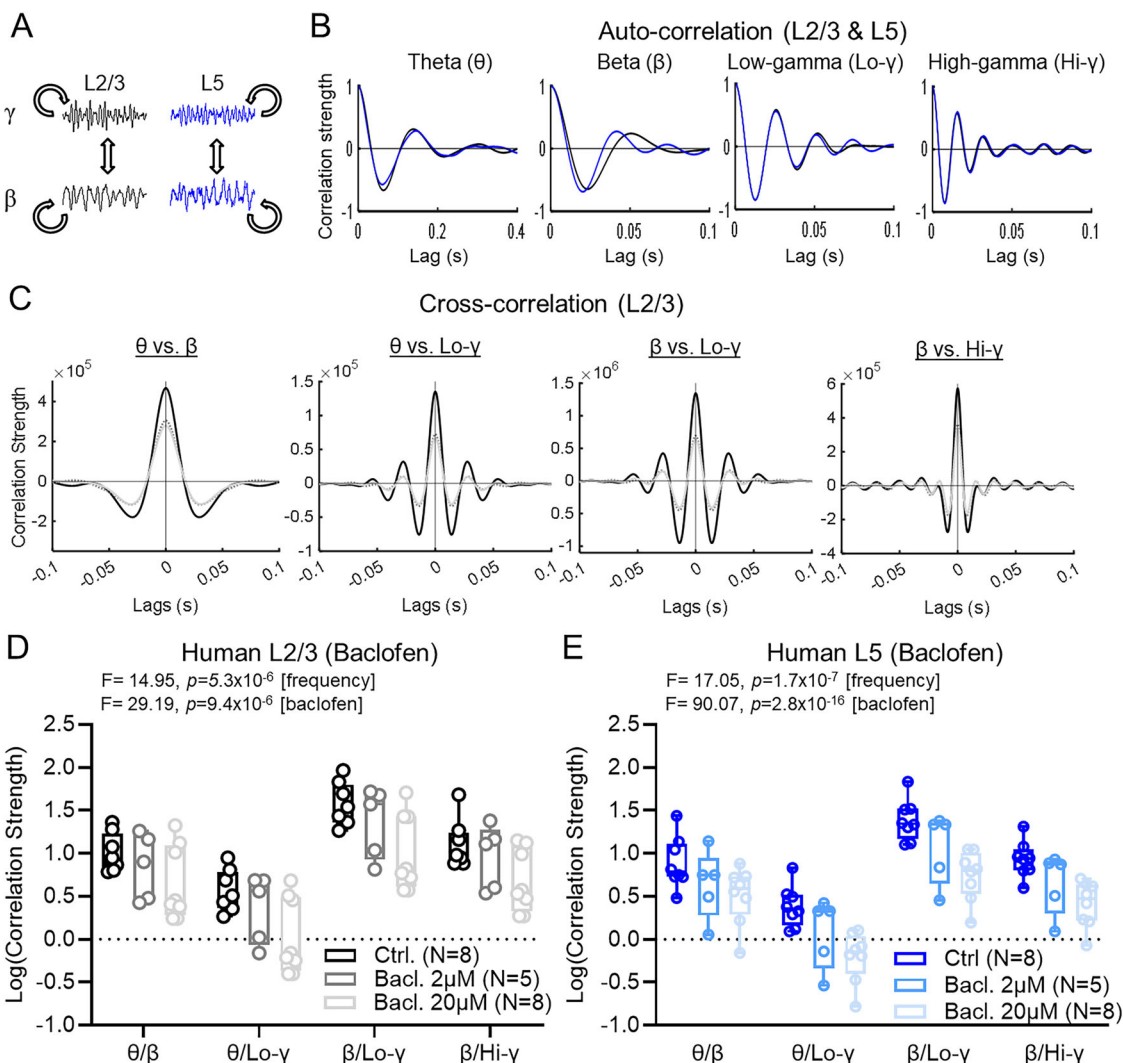

**Fig. 7 | GABA$_B$R activation de-synchronises beta/gamma coupling in human cortical circuits. A** Schematic of auto-correlation and cross-correlation analysis strategy. **B** Example plots of auto-correlation strength as a function of lag-time (ms) in LFP recordings from L2/3 (black) and L5 (blue) of the human cortex under control conditions (Kainic Acid, KA [0.6–2 μM] + Carbachol, CCh [60–200 μM]). **C** Example plots of cross-correlation strength between prominent oscillations in LFP recording from human L2/3 under control conditions (black) and following 2 μM (dark grey) or 20 μM (light grey) baclofen bath application. **D** Comparison of log-transformed correlation strength at zero lag (no time delay) between key oscillatory bands in L2/3 revealed significant frequency and baclofen effects. **E** Similar observations on cross-correlation strength were observed in human cortex L5 with respect to frequency and baclofen application. Data are shown as boxplots (25–75% range) with the median indicated and whiskers indicating maximum/minimum ranges; with individual values overlain. Statistical tests were performed with two-way ANOVA (two-sided). Source data are provided as a Source Data file.

display greater sensitivity of circuit activity to baclofen application than control patients irrespective of oscillatory frequency (Fig. 8H).

These data show that LEV boosts endogenous GABA$_B$R signalling, through a mechanism that occurs over long (>3 h) timescales. Increased GABA$_B$R signalling, while not modifying the baclofen sensitivity of neuronal oscillations, may underlie the anti-seizure properties of LEV.

## Discussion

In this study we provide the first quantitative assessment of functional GABA$_B$R control of human brain circuits, directly comparing this to that of adult and juvenile rats. We show that in adult human brain slices, postsynaptic GABA$_B$Rs produce robust currents in L2/3 and L5 PCs which are largely stable over adult life. In contrast, GABA$_B$R-mediated currents in rodent neurons displayed a rapid age-dependent decline from adolescence into adulthood. Conversely, presynaptic GABA$_B$R-mediated inhibition is largely stable across

rodent development but is weaker than human cortical circuits. These functional GABA$_B$Rs control both the strength and timing of neuronal oscillations in living human brain circuits, which are potentially more sensitive to the modulation of presynaptic GABA$_B$Rs when compared to adult rodent circuits. Finally, we show that the anti-seizure medication LEV boosts endogenous GABA$_B$R levels, in a manner that appears to be independent of seizure status. These data provide a mechanism by which GABA$_B$Rs contribute to the long-term prevention of seizure activity and epilepsy in human brain circuits.

### Differential GABA$_B$R-mediated control of neuronal excitability across life-span and species

Our data shows that human principal neurons possess strong and reliable postsynaptic GABA$_B$R signalling, regardless of location within the cortical column. Our data extends earlier studies that confirmed the presence of GABA$_B$Rs in adult human cortical tissue[3,35]. We significantly expand these findings, by performing comparative analysis

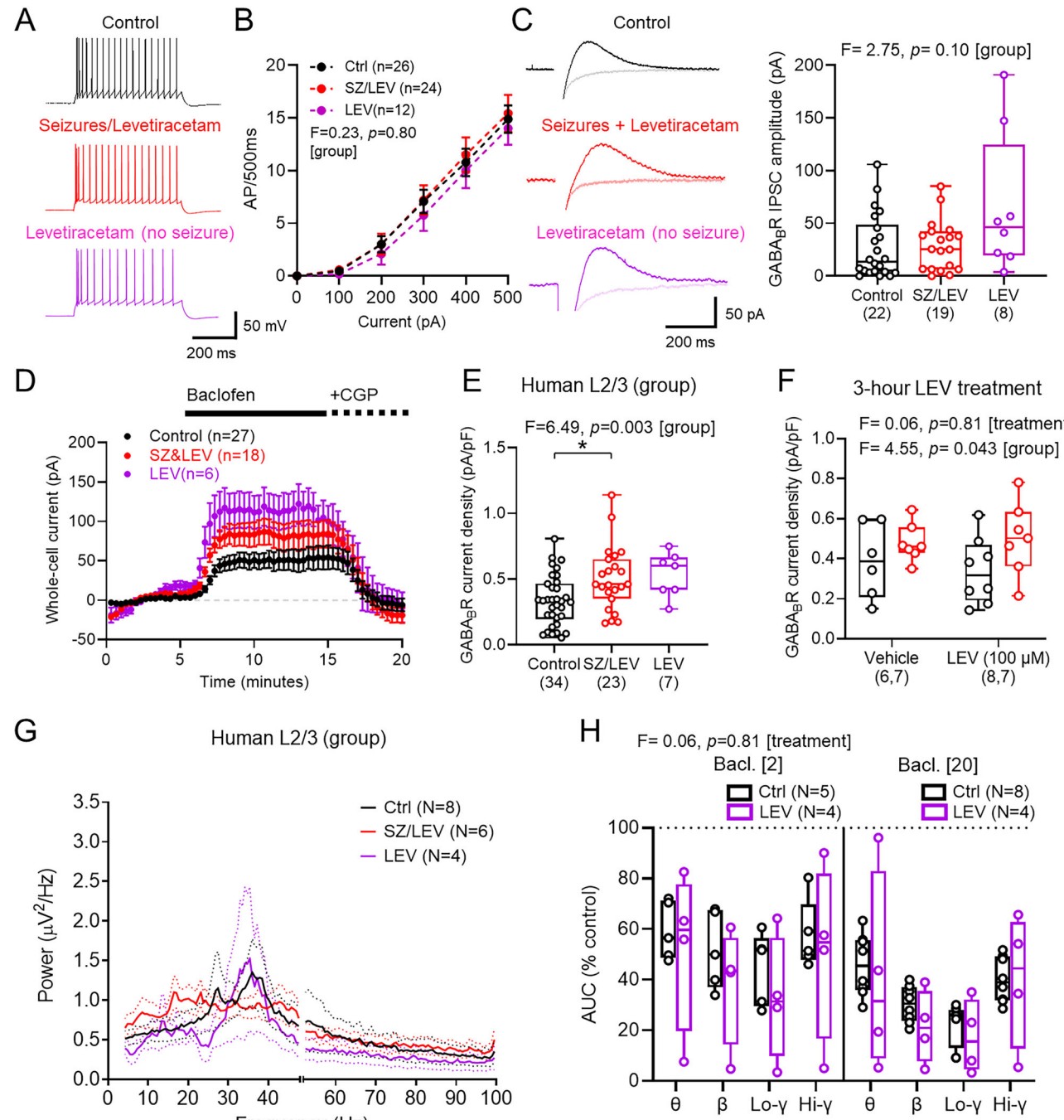

**Fig. 8 | Pre-surgical LEV leads to elevated GABA_BR signalling in L2/3 of the human neocortex. A** Responses of human L2/3 PCs to depolarising current (500 pA, 500 ms), from seizure-free patients (black), patients with seizures and received LEV (red, SZ&LEV), or patients not experiencing seizures, but receiving LEV (purple). **B** Current–frequency responses (0–500 pA, 500 ms), from control (24 cells/15 cases), SZ&LEV (24 cells/11 cases), and LEV (13 cells/4 cases). **C** GABA_BR IPSCs evoked in L2/3 PCs from control (22 cells/13 cases), SZ&LEV (19 cells/10 cases), and LEV (8 cells/4 cases), superimposed response in CGP (light traces) and IPSC amplitudes for each patient group. **D** Whole-cell current following baclofen and CGP wash-in for control (27 cells/17 cases), SZ&LEV (18 cells/11 cases), and LEV (6 cells/4 cases) patients. **E** Baclofen current-density for control (34 cells/18 cases), SZ&LEV (23 cells/13 cases), and LEV (7 cells/4 cases) patients (Ctrl vs. SZ&LEV:

$p = 0.014$; Ctrl vs. LEV-alone: $p = 0.048$; SZ&LEV vs. LEV-alone: $p = 0.705$; Tukey post hoc tests). **F** Baclofen current-density in slices pre-treated for 3 h with 100 μM LEV or vehicle, from control (Veh: 6 cells/4 cases; LEV: 8 cells/5 cases) and SZ&LEV patients (Veh: 7 cells/5 cases; LEV: 7 cells/4 cases). **G** Power spectra from L2/3 LFP recordings in KA+CCh from control (8 slices/8 cases), SZ&LEV (6 slices/6 cases), and LEV (4 slices/4 cases) patients. **H** Relative change in AUC from LEV-only patients compared to control following 2 μM (left) or 20 μM (right) baclofen wash-in. All data is shown as either mean ± SEM (**B, D, G**) or boxplots (25–75% range) with the median indicated and whiskers indicating maximum/minimum ranges (**C, E, F, H**) with individual values overlain. Statistics from LMM (**C**), 1-way ANOVA (**E**) with Holm–Sidak tests, 2-way (2-sided, **B, F**), or 3-way ANOVA (**H**).

with rodents, across layers, and with respect to clinical state. This comparison shows that L5 PCs in juvenile rats largely lack measurable GABA_BR currents at the soma, which agrees with earlier studies in young rats (<2 months old) that GABA_BRs are confined to the distal

dendritic tuft[24] and contribute to inter-hemispheric inhibition of neuronal activity[26,58]. While such specific dendritic processes are likely to be maintained into adulthood, the output of cortical columns from L5 is likely to be under stricter control of slow inhibitory signalling as we

age. High somatic labelling for GABA$_{B1}$ subunits in human cortex may reflect sequestration of the protein in the endoplasmic reticulum, perhaps due to imbalanced expression of the GABA$_{B2}$ isoform[21]. Such a mismatch in expression has been indicated to normalise whole-cell GABA$_B$R-mediated currents when measured in hippocampal interneurons[59], which may further reflect species and age-dependent differences. Further evidence for age-dependent changes in GABA$_B$R expression is scant, however, it is known that baclofen has an age-dependent effect on tonic-clonic seizures in rats[60], which may provide evidence of such changes. We cannot exclude the possibility that changes in functional GABA$_B$R postsynaptic currents arise not from the divergent expression of the receptor itself, but from its effector Kir3 (GIRK) channels. Indeed, Kir3 subunit expression matures over the first 2 months of life[61], which could also explain differences in functional currents over this period.

GABA$_B$R-mediated presynaptic inhibition of glutamate release was strongest in human tissue compared to rats at any age tested. In line with the role of presynaptic GABA$_B$Rs in controlling circuit[20] and seizure activity[45,62], we observed that low (likely presynaptic selective) concentrations of baclofen reduced the power of oscillations in human and rat brain slices. Together, these data strongly suggest that glutamatergic synapses in human neurons are under greater presynaptic inhibitory control. Given that lower concentrations of baclofen are required for presynaptic activation, this indicates that clinical administration of low-dose baclofen may have central effects, including sedation[63] and impaired memory[64]. In rodents, electrical stimulation suppressed KA-induced gamma oscillations on a timescale matching GABA$_B$R-mediated slow IPSPs[29], suggesting that the pro-cognitive effects of GABA$_B$R antagonism may be due to suppression of gamma oscillations[65,66]. Such a strong pre- and postsynaptic function of GABA$_B$Rs to the perisomatic domains of L2/3 and L5 PCs in humans could explain why GABA$_B$R modulation has enhanced effects on learning and memory as we progress through life[65].

We show that while L1/2 stimulation produced GABA$_B$R-mediated IPSCs in rats and humans and perisomatic GABA$_B$Rs exert strong functional effects on neuronal AP discharge. A number of studies have confirmed that GABAergic terminals in upper cortical layers can activate distal dendritic GABA$_B$Rs[20,24,27,58,67,68]. However, these likely minimally contribute to perisomatic inhibition in deep L2/3 and L5. The source of GABA for these receptors remains elusive but likely originates from the spillover of GABA from various synapses[18–20,67] leading to entrainment of PCs to local activity. Indeed, we show compelling evidence that GABA$_B$Rs modulate cortical oscillations, where circuit-wide activation of GABA$_B$Rs reduces neuronal discharge and desynchronises oscillatory activity. Beyond fast neuronal oscillations, high densities of GABA$_B$Rs on L5 neurons could contribute to the timing of slow-wave oscillations and the up-down states of these neurons[69].

## The role of GABA$_B$Rs contributing to human brain health and pathology

GABAergic neurotransmission has long been linked to the onset and development of seizure disorders, albeit the role of ionotropic GABA$_A$Rs displays complex and paradoxical effects on seizure propagation[70,71], with many anti-seizure medications targeting these receptors and GABA release[72]. GABA$_B$Rs have received much less attention as potential therapeutic targets for seizures, due to the fact that the agonist baclofen can exacerbate seizures[73], and antagonists have bidirectional effects[74]. This complexity is likely due to potent GABA$_B$R signalling in both excitatory and inhibitory neurons alike, leading to shifts in excitatory/inhibitory balance that may produce complex circuit effects (reviewed in Kulik, et al.[16]).

GABA$_B$Rs themselves have long been implicated in seizure disorders, including temporal lobe epilepsy (TLE), where expression of receptor subunits has been suggested to be elevated in post-mortem human brain[2,4], but with impaired function in neurons recorded from

the seizure focus[3]. This is mimicked in rodent models of TLE, where rapid loss of GABA$_B$R subunits was observed in the dorsal hippocampus[6]. Our data does not address whether GABA$_B$Rs are lost following seizures, however, we show that in patients who have received LEV functional GABA$_B$R currents are elevated, irrespective of seizure history. The mechanism by which LEV mediates long-term suppression of seizures is not fully understood. It is known that it directly binds to Synaptic Vesicle Protein 2A (SV2A)[75], which is present in axon terminals of both glutamatergic and GABAergic neurons and may be strongly expressed in basket cell axon terminals[76]. Indeed, administration of LEV leads to increased GABA levels in the brain[77], which agrees with a central role at GABAergic synapses. The mechanisms by which LEV leads to elevated GABA$_B$R currents remain to be determined. However, this is likely a specific effect, as previous studies of GABA$_A$R signalling in resected human brain tissue did not observe an effect on ionotropic currents[78]. The ability to regulate endogenous GABA$_B$R functional currents represents a unique and promising avenue for the development of future anti-seizure medications, which may lack many of the deleterious side effects of LEV[79,80]. Intriguingly, we observed that baclofen did not produce more pronounced effects on neuronal oscillations in LEV-only patients. This may reflect a compensation to altered GABA release probability that has yet to be identified, or indeed that the effects of baclofen on post-synaptic membranes are saturating at the concentrations used in our study. The absence of an altered response to 2 µM baclofen indicates that presynaptic GABA$_B$R function may be unaffected in LEV patients. This may indicate that effects on GABA$_B$R signalling are confined to postsynaptic membranes of excitatory neurons, and not inhibitory interneurons, but this requires further investigation.

## Technical considerations

Given the nature of the recordings we have performed, there are several key technical considerations. First, all intracellular recordings performed in the current study were made from the cell body. This could overlook the role of GABA$_B$Rs in the distal dendrites, due to electrotonic compartmentalisation, particularly for L5 neurons[24,58,81]. While we fully appreciate the importance of dendritic GABA$_B$Rs, particularly in the case of integration of synaptic inputs, the purpose of this study was to assess the impact on the spiking ability of neurons in response to activity within their layer. Determining how GABA$_B$Rs contribute to dendritic computation in human neurons is a worthwhile undertaking, especially given differences in dendritic arborisation and electrotonic propagation[37], but is out of the scope of this current study.

Second, our human data has been collected from a relatively old patient cohort (median age = 57.5 years). We sought to address this age of patients by performing parallel experiments in adult rats, but there still remains a clear chronological difference in brain age between these groups[82], which is limited by the patients we can obtain brain samples from and making direct comparisons complicated. Furthermore, our human cohort does not fully capture adult brain development, as our youngest sample was 28 years old. Further studies examining more adolescent or paediatric cohorts would address this issue. Nevertheless, understanding the function of GABA$_B$Rs across the adult lifespan into old age has distinct merits, in particular, as clinical trials for baclofen are often in older patient cohorts[83], GABA$_B$R expression may display age-dependent effects[38] and GABA$_B$Rs are implicated in neurodegenerative conditions (e.g. Alzheimer's disease[84]). One consideration is that, despite observing large differences between the mean baclofen responses of patients receiving LEV, we failed to observe statistical differences from the control. Given that these patients were recruited in parallel to a clinical trial, which has since ended, it is not possible to obtain further samples from them. As such future studies using preclinical models may be required to confirm the role of LEV alone on GABA$_B$R function.

In conclusion, we provide the first quantitative analysis of functional GABA$_B$R-mediated currents and their role in controlling cellular and circuit-level excitability in the adult human cortex. We show that humans display phylogenetic differences in the functional control of neuronal activity at both presynaptic and postsynaptic GABA$_B$Rs, which efficiently control local circuit activity, and are crucially involved in the mechanism of action of the anti-seizure medication levetiracetam. These data highlight the importance of GABA$_B$R signalling mechanisms over the lifespan of humans, particularly with respect to epilepsy research and the generation of anti-seizure medications.

## Methods

### Animals

In vitro electrophysiological experiments were performed in acute slices from either 1-month (28–38-day)-old, 6–8-month-old, or 12–14-month-old male Long-Evans Hooded rats of both sexes. All experiments were performed in accordance with institutional (University of Edinburgh, UK) and UK Home Office guidelines (ASPA: PPL: P2262369). All experiments were approved by the Named Veterinary Surgeon or their delegate, under the guidance of the Animal Welfare and Ethical Review Body (AWERB), University of Edinburgh. All rats were kept on 12-h light/dark cycles, housed in cages of 2-5 rats, and given ad libitum access to food and water.

### Resected human tissue collection

All human brain tissue collection was subject to local and regional ethical approval (NHS Lothian: REC number: 15/ES/0094, IRAS number: 165488; NHS Newcastle: IRAS 173990). Prior to tissue collection, all patients provided written consent for access to tissue and anonymised patient information usage (Edinburgh: NHS Lothian Caldicott Guardian Approval Number: CRD19080). No patient identifying information was stored and all patient information anonymised. Human neocortical brain tissue was collected during the resection of brain tumours, of which 32% had glioma, 47% had glioblastoma, 13% had brain metastases, and 8% had other tumours. Based on our data, we identified 3 key human groups: control individuals (no seizure history or anti-seizure medication), recent seizure history and pre-operative levetiracetam, and patients who had received levetiracetam (but who had not experienced seizures). This latter group had been prescribed prophylactic levetiracetam 500 mg twice daily, for at least 3 days prior to surgery[85], as part of a now-closed clinical trial (EudraCT Number: 2018-001312-30). All patient's tissue was recorded and analysed blind to the group. Patient attributes are listed in Supplementary Table 1.

Brain tissue was collected and sliced[86,87]. Briefly, following the surgical opening of the skull and dura, a small piece of cortical tissue was resected en route to accessing more deeply sited tumours. This brain tissue was then rapidly transferred to ice-cold, oxygenated HEPES modified artificial cerebrospinal fluid (HEPES-ACSF; in mM: 87 NaCl, 2.5 KCl, 10 HEPES, 1.25 NaH$_2$PO$_4$, 25 Glucose, 90 Sucrose, 1 Na$_2$-Pyruvate, 1 Na$_2$-Ascorbate, 7 MgCl$_2$, and 0.5 CaCl$_2$; pH 7.35 with NaOH) and then transported to the laboratory (ca. 20–40 min). Brain tissue was then mounted in 3% agar gel (nominally 30 °C), blocked and mounted in an oscillating blade vibratome (VT1200S, Leica, Germany). For whole-cell recordings, 300 µm-thick slices of cortex were cut and transferred to a submerged storage chamber containing sucrose-modified ACSF (in mM: 87 NaCl, 2.5 KCl, 25 NaHCO$_3$, 1.25 NaH$_2$PO$_4$, 25 glucose, 75 sucrose, 7 MgCl$_2$, 0.5 CaCl$_2$, 1 Na$_2$-Pyruvate, 1 Na$_2$-Ascorbate) warmed to 35 °C and bubbled with carbogen (95% O$_2$/5% CO$_2$) for 30 min then placed at room temperature. For extracellular recordings, 500 µm-thick slices were cut and stored in a liquid/gas interface chamber containing recording ACSF (in mM: 125 NaCl, 2.5 KCl, 25 NaHCO$_3$, 1.25 NaH$_2$PO$_4$, 25 glucose, 1 MgCl$_2$, 2 CaCl$_2$); bubbled with carbogen and maintained at room temperature.

### Rat brain slice preparation

Rat brain slices containing primary somatosensory cortex were prepared[88]. Briefly, rats were sedated with isoflurane, anaesthetised with sodium pentobarbital, then transcardial perfusion was performed with carbogenated (95%O$_2$/5% CO$_2$) ice-cold sucrose-ACSF. Following perfusion rats were decapitated, and their brain rapidly removed into ice-cold, carbogenated sucrose-ACSF. For whole-cell recordings, 400 µm-thick coronal brain slices were cut on a Vibratome (VT1200s, Leica, Germany) in semi-frozen sucrose-ACSF, then stored submerged in sucrose-ACSF warmed to 35 °C for 30 min and subsequently at room temperature. For extracellular oscillation analysis, 500 µm-thick slices were cut and stored in a liquid/gas interface chamber containing recording ACSF; bubbled with carbogen, and maintained at room temperature.

### Intracellular recordings

For whole-cell patch-clamp recordings, slices were transferred to a submerged recording chamber which was perfused with carbogenated recording ACSF (in mM: 125 NaCl, 2.5 KCl, 25 NaHCO$_3$, 1.25 NaH$_2$PO$_4$, 25 glucose, 1 MgCl$_2$, 2 CaCl$_2$) at 5–6 mL/min at 31 ± 1 °C by an inline heater. Slices were visualised under Köhler illumination by means of an upright microscope (Slicescope, Scientifica, UK), equipped with a ×40 water-immersion objective lens (NA 0.8; Olympus). Whole-cell patch-clamp recordings were accomplished using a Multiclamp 700B amplifier (Molecular Devices, CA, USA), as previously described Oliveira et al.[89]. Recording pipettes were pulled from borosilicate glass capillaries (1.5 mm outer/0.86 mm inner diameter, Harvard Apparatus, UK) on a horizontal electrode puller (P-97 or P-1000, Sutter Instruments, CA, USA). When filled with intracellular solution (in mM: 142 K-gluconate, 4 KCl, 0.5 EGTA, 10 HEPES, 2 MgCl$_2$, 2 Na$_2$-ATP, 0.3 Na$_2$-GTP, 10 Na-phosphocreatine, 0.1% Biocytin, corrected to pH 7.4 with KOH, 295–305 mOsm) a pipette resistance of 2–6 MΩ was achieved. Unless otherwise stated, all voltage-clamp recordings were performed at a holding potential of −65 mV, and all current-clamp recordings from the resting membrane potential ($V_M$). For all recordings series resistance ($R_S$) was monitored but not compensated in voltage-clamp and the bridge balanced following pipette-capacitance compensation in current-clamp. Signals were filtered online at 2–10 kHz using the built-in 2-pole Bessel filter of the amplifier, digitised and acquired at 20 kHz (Digidata 1550B, Axon Instruments, USA), using pClamp 10 (Molecular Devices, CA, USA). Data was analysed offline using the open-source Stimfit software package [90] http://www.stimfit.org. The liquid junction potential was measured as −12 mV and measurements were not adjusted.

In whole-cell recordings, the intrinsic properties of recorded neurons were characterised by current-clamp from resting membrane potential. A family of 500 ms hyper- to depolarising current steps (−250 to +250, 50 pA steps; or −500 to +500, 100 pA steps) were used depending on the initial input resistance of the neuron. Cells were identified on the basis of the voltage response and the resulting train of action potentials (AP) elicited by a family of hyper- to depolarising current steps (50 pA, 500 ms duration; −500 to +500 pA). Neurons were rejected from further analysis if resting membrane potential was more depolarised than −50 mV, APs failed to overshoot 0 mV, initial access resistance ($R_A$) exceeded 30 MΩ, or $R_A$ fluctuated by >20% over the time course of the experiment.

### Characterisation of postsynaptic GABA$_B$R-mediated effects

To identify GABA$_B$R-mediated currents, neurons were recorded in the presence of ionotropic receptor blockers, CNQX or NBQX (10 µM), DL-AP5 (50 µM), and either picrotoxin (50 µM) or SR-95531 (Gabazine, 10 µM), which were bath applied. Extracellular stimuli were delivered via a bipolar twisted Ni:Chrome wire electrode placed at the border of L1 and L2. GABA$_B$R-mediated IPSCs were evoked by 200 Hz trains of 5 stimuli[18,28] and a minimum of 15 IPSCs were collected. The amplitude

of GABA$_B$R-mediated IPSCs was measured in average traces (>5 traces) over a 10 ms peak, within a 200 ms region relative to the pre-stimulus baseline. GABA$_B$R-mediated whole-cell currents ($I_{WC}$) were assessed by 5–10-min bath application of the canonical agonist R-baclofen (10 μM). Specifically, $I_{WC}$ change for drug wash-in was measured as the holding current required to maintain the recorded cell at −65 mV voltage-clamp. Peak baclofen response was measured as the maximum change in $I_{WC}$ over a 1-min window of each pharmacological epoch following subtraction of baseline holding-current. For density normalisation, the peak baclofen $I_{WC}$ was divided by the membrane capacitance derived from the input resistance and membrane time constant from hyper-polarising current steps. To confirm that baclofen-induced $I_{WC}$ was GABA$_B$R mediated, the potent and selective antagonist CGP 55,845 (CGP, 5 μM) was applied to the bath.

### Pharmacological manipulation of neuronal discharge

For baclofen puff experiments, a patch pipette was filled with 150 mM NaCl containing 50 μM baclofen and lowered to the slice adjacent to the recorded cell. Upon successful establishment of a whole-cell recording, in the presence of NBQX (10 μM), DL-AP5 (50 μM) and either picrotoxin (50 μM) or SR-95531 (Gabazine, 10 μM), the puff pipette was placed just below the surface of the slice, ~50–100 μm distal from the recorded cell. Based on the intrinsic characterisation of the cell (see above), the rheobase was identified and a +25 pA bias current was applied for baclofen puffs. Each puff was delivered via a PicoSpritzer (10 mbar, 200 ms) at 1-min intervals. In most cells, 1–3 baclofen puffs were applied under voltage-clamp (−65 mV) to determine the current amplitude evoked. Following successful measurement of AP discharge following baclofen puffs, CGP was applied to the bath to confirm that inhibition was baclofen mediated. For analysis, spontaneous APs detected and frequency calculated in 200 ms temporal bins, plotted against time. The degree of perisomatic inhibition was calculated 2–3 s after the onset of puff application.

### Characterisation of presynaptic GABA$_B$R function

Monosynaptic EPSC were examined in human and rat L2/3 and L5 neurons in the presence of picrotoxin (50 μM) added to the perfusing ACSF. To evoke synaptic responses, extracellular stimuli were delivered via a Ni:Chrome twisted bipolar electrode placed in either L1 and L4 (for L2/3 PCs) or L1 and L5 (for L5 PCs). EPSCs were elicited using paired stimulus (2× stimuli at 20 Hz) repeated at 0.1 Hz, with responses to L1 and L4 or L5 stimulation interleaved. Stimulus intensity was titrated to produce a monosynaptic response of ~100 pA. After 5 min of stable recording, the GABA$_B$R agonist baclofen was applied to the bath at 10 μM. Following steady state of baclofen wash-in, we then applied the potent and selective GABA$_B$R antagonist CGP-55,845 (5 μM) to confirm receptor specificity. The amplitude of EPSCs was measured over a 10 ms window following the stimulus artefact and mean data presented as the average of 10 traces normalised to baseline levels over the 2 min prior to baclofen wash-in. For PPR comparisons, the second EPSC amplitudes were divided by the first.

### In vitro oscillation experiments

For local field potential (LFP) recordings, slices were transferred to an interface recording chamber which was perfused with carbogenated recording ACSF at 2–3 ml/min and maintained at 30 ± 1 °C. Recording pipettes with a resistance of 1–3 MΩ were pulled from borosilicate glass capillaries (1.5 mm outer/0.86 mm inner diameter, Harvard Apparatus, UK) on a horizontal electrode puller and filled with recording ACSF. Slices were visualised by means of a wide-field microscope (Leica, Germany) and pipettes placed in L2/3 and L5. To induce population-level oscillations, kainate (KA, 0.6–2 μM) and car-bachol (CCh, 60–200 μM) were applied to perfusing ACSF for a minimum of 30 minutes. Baclofen (2 and 20 μM) or CGP (5 μM) were applied to the perfusing ACSF, in addition to KA and CCh. LFP

recordings were accomplished using a 2-channel amplifier (EXT-02 B, NPI Electronics, Germany). Signals were low-pass (1 Hz) and high-pass (500 Hz) filtered online, and acquired at 20 kHz (Micro1401-3, Cambridge Electronic Design, UK). All raw data were collected using Spike2 (Cambridge Electronic Design, UK). For analysis, 2-min epochs were notch filtered at (49–51 Hz), bandpass filtered from 1 to 100 Hz, and then fast-Fourier transform analysis was performed to generate power spectra. Signals were rejected from analysis if epileptiform-like activity was present following induction of oscillatory activity or if beta/gamma (12–100 Hz) power was lower than 0.4 μV$^2$.

### LFP/AUC analysis

Fast-Fourier transforms (FFTs) were performed on 2-min epochs of the filtered signal to determine the amplitude (μV$^2$) and frequency (Hz) of each signal following 30–40 min KA+CCh, 2 and 20 μM baclofen application. Employing a resolution of 0.61 Hz within a Hanning window, FFT size generated 164 data points spanning 0.61–99.4 Hz, capturing power and corresponding frequency. FFT amplitude values were normalised to the frequency resolution (0.61 Hz) yielding a power spectral density plot illustrating amplitude as a function of frequency (μV$^2$/Hz). Frequency bands were defined as theta (4–10 Hz), beta (10–30 Hz), low-gamma (30–50 Hz), and high-gamma (50–85 Hz). Area under the curve (AUC) was calculated as the amplitude of each frequency band between lower and upper bandwidths during control (KA+CCh) or baclofen conditions. AUC values were normalised to KA +CCh to quantify amplitude changes post-baclofen application.

### Auto- and cross-correlation analysis

The same 2-minute epochs used in the spectral analysis were imported into MATLAB and analysed using custom MATLAB scripts. Signals were subdivided into three 10-s epochs (0–10, 50–60, and 110–120 s) then bandstop filtered across the four distinct frequency bands. Auto-correlation used the Xcorr() function to determine the correlation strength of each frequency band over time by comparing a static signal to a time-lagged copy of itself. Cross-correlation similarly used the Xcorr() function to gauge the strength between prominent frequency pairs following KA+CCh and baclofen conditions in all L2/3 and L5 recordings. Cross-correlation between distinct frequency bands was measured from slices with simultaneously oscillating L2/3 and L5 recordings. Representative correlograms were generated to produce a visualisation of the coupling between oscillations. The strength of cross-correlation analysis was quantified using the peak value at zero-time lags.

### Phase–amplitude coupling analysis

The strength of phase–amplitude coupling (PAC) between distinct frequency bands was measured using a custom MATLAB script, based on earlier studies[48]. 2-min epochs of 50 Hz bandpass filtered signal were imported into MATLAB. Signals were zero-phase filtered using the filtfilt() function, then bandstop filtered to isolate each distinct frequency band. Hilbert transforms were run to generate a normalised phase-amplitude plot of the isolated lower frequency phase angles and the amplitude envelope of the higher frequency oscillation. The modulation index measure was used to quantify the PAC strength, calculating the deviations of phase-amplitude from a uniform distribution, with a smaller variation producing a larger modulation index value indicative of a stronger PAC.

### Visualisation of recorded neurons

Following successful experiments, recorded cells were sealed with outside-out patch formation and then fixed in 4% paraformaldehyde (PFA) diluted in 0.1 M phosphate buffer (PB) for 24–48 h at 4 °C[91]. Slices were rinsed in PB, then phosphate-buffered saline (PBS; 0.1 M PB + 0.9% NaCl). Slices were then incubated for 48–72 h in a PBS solution containing 0.3–0.5% TritonX-100 and 0.05% NaN$_3$ and streptavidin

conjugated to either AlexaFluor568 or AlexaFluor633 (1:500; Invitrogen, UK). Slices were then washed in PBS, then PB, and mounted on glass slides with a hard-set mounting medium (Vectashield, Vector Labs, UK) and then cover-slipped.

## Immunohistochemistry

Immunocytochemistry was performed on cortical tissue from perfusion-fixed rat brains (4% PFA + 0.1 M PB) or from immersion fixed pieces of human brain (24 h at 4 °C) after fixation tissue was transferred to PBS. Tissue was equilibrated with 20% sucrose and 30% glycerol in PBS and stored at −20 °C until sectioned. Tissue blocks were cryoprotected overnight in 30% sucrose in PBS, then 50 µm-thick sections were cut on a freezing microtome. Sections were washed in PBS and then blocked for 1 h at room temperature in 10% normal goat serum (NGS), 0.3% TritonX-100 and 0.05% $NaN_3$ in PBS. Sections were incubated in primary antibodies raised against $GABA_{B1}$ (mouse, 1:400, N93A/49, NeuroMab, UNC Davis, CA, USA) and NeuN (rabbit, 1:500, ABN78, Millipore) diluted in PBS containing 5% NGS, 0.3% TritonX-100 and 0.05% $NaN_3$, for 24–72 h at 4 °C. Sections were washed in PBS, then secondary antibodies (anti-mouse and anti-rabbit AlexaFluor488 and AlexaFluor 568, Invitrogen, UK) were diluted in PBS containing 3% NGS, 0.1% TritonX-100 and 0.05% $NaN_3$, for 3 h at room temperature or 24 h at 4 °C. Slices were rinsed in PBS, then PB and mounted on glass slides with Vectashield hard-set mounting medium (H1400, Vector Labs, UK.

## Imaging and image analysis

Biocytin-filled cells and immunostained sections were imaged with a laser scanning confocal microscope (SP8, Leica, Germany) using a ×20 objective and z-axis stacks of images collected (2048x2048 pixel radial resolution, 1 µm axial steps). Example cells were reconstructed offline from image stacks digitally stitched and segmented using semi-automatic analysis software (Simple Neurite Tracer plug-in for the FIJI software package (http://fiji.org)[92]. For $GABA_{B1}$ immunolabelling, sections were images at ×20 magnification, (2048 × 2048 pixel radial resolution, 1 µm axial steps) with z-stacks of the full 50 µm slice taken to span the cortical column. Images were stitched offline, and then the fluorescent intensity of $GABA_{B1}$ was measured along a 100 µm wide region of interest through the cortical column from z-projected images (15 µm projection). Arbitrary fluorescent intensity was measured with the "plot profile" function, which was background subtracted (white matter as background), then z-scored, and the position from pia normalised and data binned within this range (pia = 0, L6/white matter border = 1).

## Statistical analysis

All statistical comparisons were made using either linear mixed-effects models (LMM; Figs. 1, 2, 3, 5, and 8) to minimise potential pseudoreplication[93], Mann–Whitney tests (Fig. 4), 2- and 3-way ANOVAs (Figs. 6 and 7), and linear regression (Pearson's tests, Fig. 2). Post hoc analysis was performed using Tukey post-tests, adjusted for multiple comparisons. For LMM analysis, residual plots were generated and data were tested for normality using the Shapiro–Wilk test. If required, data were transformed using either log, square-root (for data with zero values) or Box Cox; in that order. LMM analysis was then performed using the "lmer" function in R-studio. Age and case/animal were assigned as random effects, where appropriate; then type III ANOVA was performed on the model to assess main effects (e.g. age, species, layer, clinical group). If a statistical interaction between main effects was observed, then post-hoc analysis was performed (Tukey post hoc tests). For all comparisons data is shown as box-plots with median and 25–75% percentiles displayed in the box; minimum and maximum extents are shown as whiskers, with individual data points shown from cell averages, with the exception of immunohistochemistry and LFP analysis where data from 1 slice per case/animal is shown. Given the near-random recruitment of patients, we could not assure

biological sex balance in our human data, hence we did not perform analyses comparing sex of individuals, but this metadata is available in Supplementary Table 1. All statistical analysis was performed using GraphPad Prism 10 or R with R-studio. For current-frequency, time-course, and spectral plots, the mean ± SEM is displayed. All tables state the mean ± SD of observed data. Statistical significance was assumed if $P < 0.05$. A full summary of all reported values is provided in the supplementary materials.

## Reporting summary

Further information on research design is available in the Nature Portfolio Reporting Summary linked to this article.

## Data availability

Additional information is available upon request. Source data are provided with this paper.

## Code availability

All code used in this study is available in supplementary materials.

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

## Acknowledgements

Most of all we would like to thank the patients who generously agreed to donate tissue for this study, NHS Lothian NRS BioResource and Tissue Governance Unit, and the EMERGE Research Nurse team, in particular Allan MacRaild, Ikeoluwa Adekoya, Anuka Boldbaatar and Sarah Risbridger. Thanks are given to David J.A. Wyllie and Peter C. Kind for their support in conducting this work. We also thank Naima Elose Borras for help with the initial image analysis; Kind Lab members for helpful discussions; and Colin Smith, Mark Cunningham, and Giles Hardingham for initial discussions around establishing tissue collection. Finally, this project was funded through generous support from the Simons Initiative for the Developing Brain (S.A.B. and M.W.), Edinburgh Neuroscience Neuroresearchers Fund (S.A.B.), RS McDonald Seedcorn Fund (S.A.B.), Medical Research Council UK (S.A.B.—MR/Y014529/1), Race Against Dementia (C.D.—ARUK-RADF-2019a-001), The James Dyson Foundation (C.D.), Alzheimer's Society UK (C.D.—AS-PG-21-006).

## Author contributions

M.W.: Investigation, methodology, software, formal analysis; visualisation, writing—original draft, writing—review and editing; A.S.: Formal analysis, writing—reviewing and editing; L.T.: Resources, writing—review and editing; S.M.: Resources, writing—review and editing; R.M.: Resources, writing—review and editing; T.M.: Resources, writing—review and editing; A.J.: Resources; C.C.: Resources; F.L.: Resources and project administration; I.L.: Resources; C.D.: Resources, project

administration, writing—review and editing; P.B.: Resources, methodology, project administration, writing—review and editing; S.A.B.: Conceptualisation, methodology, validation, formal analysis, visualisation, writing—original draft, writing—review and editing, project administration, and funding acquisition.

## Competing interests

The authors declare no competing interests.
