## [Transparent Peer Review file · Nature Communications]

Phylogenetic divergence of GABAB receptor signaling in neocortical networks over adult life

Corresponding Author: Dr Sam Booker

Version 0:

Reviewer comments:

Reviewer #1

(Remarks to the Author)

Wilson and co-authors studied GABA_BR action on human and rat neocortical network using electrophysiological methods (whole-cell patch-clamp and local field potentials in vitro). They showed some similarities and differences between rats and human in terms of the modulation of the neocortex through GABA_BRs. They claim that the strength of postsynaptic GABA_BR inhibition declines over lifespan in rodents but this inhibition is stable in human over life. The effect of presynaptic GABA_BRs is more profound in human than rats. GABA_BRs control network oscillations in human. Finally, they showed that the application of anti-seizure-medication levetiracetam for a few days before the tumor surgery enhances GABA_BR inhibition in human.

These data are very valuable taking into account that the study has been done in human tissue. The manuscript should be published after revision.

Major:

- lines 30-33: In the introduction, authors should mention that % of inhibitory neurons in the neocortex is quite different in human and rats/rodents.
- Figure 1: How was done statistical analysis in Fig 1 B? There are different numbers of cells over B-C and D. Please revise it.
- Please explain in the methods what "neuronal gain" means and how it was calculated.
- What is a schematic to indicate number of cells/animals/human individuals? Please provide it to every experiment and figure.
- lines 144-45: "In human cortical neurons, we found that L5 PCs had slightly (+18%) higher baclofen current density, than L2/3 PCs ($t(28,10)=0.69$, $p=0.49$, Holm-Sidak test)."
Is this statistically different? Of not, maybe it should not be considered. Please explain it.
- Figure 2: please add some examples of the recordings for Fig.A.
- Lines 193-5: "The decay time-constants of evoked slow-IPSCs were more variable, particularly in adult rats and humans, with a tendency to longer decay times in L5 neurons (Figure 3D)."
Is this statistically different? Of not, maybe it should not be considered. Please explain it.
Also, it should be discussed in more details why kinetics might be different.
- Figure 5: E and F would benefit from a better figure legend on the plot. It is hard to follow all the configurations of the stimulation. Please add statistics to the panels E and F.
- Figure 6: add statistics to the panels D and E.
- Line 369: please specify that these experiments were done in human tissue.
- lines 438-444: It seems that something is lacking here, especially because the second sentence is not finished. Please correct it.
- lines 443-445: "(...) although individuals who had received LEV, but not experienced seizures tended to possess larger GABA_BR-mediated IPSCs (Figure 8C)."
Is this statistically different? Of not, maybe it should not be considered. Please explain it.
- lines 745-758: add the information about holding potential
- Why Holm-Sidak was used?
- Supplementary Table 2 and 3. There is no data about rats. Please provide it here or as a separate Table.
- Supplementary figure 2: please provide number of cells/rats.
- Supplementary figure 5: add figure legends directly above of every plot.
- Supplementary figure 9: add number of individuals and statistics.
- Supplementary Table 3: this table is not cited in the text, please do this. Also, it is unclear what data are here. Is this data

pooled from layers 2/3 and 5?. Comparing to Table 2, these data are mostly similar to L2/3 data. However, e.g. there are different values of SEM for input resistance. Maybe this is a mistake.

Minor:

-line 344: delete "in"

-line: 692: delete "either"

-line 733: add "were" after "measurements"

-lines 771-773: correct the sentence adding were/was

Reviewer #2

(Remarks to the Author)

This study uses electrophysiological analysis of neurons in neocortical brain slices from rat and freshly resected human brain tissue to investigate the properties of GABAB receptor signaling on synaptic, neuronal and neuronal network function. The data show several findings:

- there are cell type-specific developmental changes in rat cortical pyramidal neuron GABAB responses.
- human neurons appear to have GABAB-mediated responses that are differentially located in the cell bodies of layer 2/3 and layer 5 pyramidal neurons.
- pharmacological activation of GABAB receptors can alter neuronal and neuronal network activity patterns.
- treatment of patients with the anti-seizure drug Levetiracetam seems to alter the amount of available GABAB-mediated signaling in human layer 2/3 pyramidal neurons.

Overall, there are some intriguing findings shown in this paper and some novel information, particularly in relation to the rather unique set of human samples. In a way, there are several strands to the data presented – developmental inter-species comparison, mechanisms of how GABAB effects are mediated, clinical significance of GABAB for epilepsy treatments. Each provides some new findings, but each is slightly incomplete. Furthermore, while the experimental quality is itself of a high standard, there are some limitations in the experimental design and therefore the insights we can make, particularly in relation to realistic activation of GABAB receptors and into some of the proposed mechanisms underlying changes in GABAB-mediated effects.

Specific points:

As acknowledged in Discussion, the youngest human samples are from ~30 years old, which is probably more equivalent to 6 month rather than 1 month old rat. So age comparisons are not really overlapping. For example, the developmental decrease in GABAB current density in rats is probably too early developmentally to be captured by the human ages sampled. An older rat age, or young human ages, would allow more directly comparisons of age-dependent changes across species. Without this, there are concerns that the inter-species comparisons, that are a key part of the first part of the paper, are not strongly underpinned.

It is important to note that changes in GABAB-driven currents could reflect changes in the downstream potassium channel expression/properties or G-protein signaling in addition or instead of changes in GABAB receptors themselves.

Throughout the data analysis, cell or slice is used as the experimental unit. This is likely a reasonable choice but there is clearly great potential, particularly in the human data, that there will be significant effects associated with the individual samples (perhaps due to case-to-case variation in tissue handling, patient demographics or clinical status). The point is made fairly that in the Methods, with low numbers of recordings from each sample, it may be difficult to measure inter-sample variance accurately. However, this information could be included in hierarchical/mixed statistical models. At the least, it is important to be clear how many samples are associated with each dataset (as well as cell/slice n numbers), either in the text or represented in the datapoint symbols.

Line 234 – it is suggested GABAB receptors preferentially localize to the cell soma with age but there is no direct measurement of receptor density there, which is the suggested mechanism. Could a direct measure of receptor density in somatic membranes be made using some immunohistological or electron microscopy approach?

Figure 5 – were the input pathways confirmed as independent via interleaved paired pulse stimuli?

What is really missing is effect of endogenous GABAB receptor activation on synaptic stimulation. When CGP is added at the end of the baclofen experiments, the EPSC size does not get bigger than baseline, so doesn't this suggest that there is no GABAB-mediated effect triggered by the synaptic stimulation. Maybe this low-frequency, single test pulse is not sufficient to trigger enough GABA release, but it does seem possible with the high-frequency, 5 pulse stimulus train used in Figure 3. It would be good to show that EPSCs evoked by such a high-frequency stimulus train are indeed altered by the presence of CGP – this would confirm an effect of synaptically-evoked GABAB receptor activation.

Likewise, in the oscillation experiments shown in Figure 6, would we not expect GABA release to be extensive during the KA/CCh-triggered oscillatory activity, and therefore be activating GABAB receptors. It would be more interesting to see any effect on oscillations of blocking GABAB receptors with CGP rather than activating them all artificially with baclofen.

Line 440 – sentence appears to have been cut off.

Lines 442-445 – "We found no overall difference in GABABR-mediated IPSC amplitudes between any patient group

(KW=4.47, P=0.107, Kruskal Wallis Test), although individuals who had received LEV, but not experienced seizures tended to possess larger GABABR-mediated IPSCs (Figure 8C).” The assertion about these larger currents seems to correspond to data from two cells (not clear if from the same patient) and is not corroborated by the stats assessment, so should be removed.

How well does 100uM LEV correspond to doses in the brains of patients? This would be discussed.

There should be discussion about how to reconcile that LEV patients have greater GABAB currents evoked by baclofen but this same treatment has no differential effect on oscillatory activity.

Version 1:

Reviewer comments:

Reviewer #1

(Remarks to the Author)

Wilson and co-authors studied GABA_BR functioning in human and rat neocortical networks using electrophysiological methods in acute slices. The experiments are elegant and performed with high standard. Studies on human tissue are always valuable. I recommend the manuscript for publication.

The authors have satisfactorily addressed my concerns.

One minor comment. The sentence should be re-written in lines 449-51: “In human neocortex, because we found low concentrations of baclofen inhibited the strength of oscillations across a broad range of frequencies, likely due to the activation of presynaptic GABA_BRs.”

Reviewer #2

(Remarks to the Author)

The authors have made an impressive response to my queries on the original manuscript. I am satisfied that they have addressed my requests and feel the manuscript is improved. As such, I think the manuscript should be published.

One additional query regarding the immunofluorescence data now shown in Figure 3A - in the representative images, the staining pattern in human tissue appears to be more heavily based on signal from cell bodies, compared to the rat tissue, which has a stronger neuropil element. If this is truly representative of the data as a whole, it would be worth adding discussion of how this might relate to the laminar profiles and the somatic current density differences.

We thank the reviewers and editors for their careful appraisal of our manuscript. In preparing the revised manuscript we have taken into account all comments and have addressed them accordingly with new figures, analysis, and discussion. We provide a point-by-point response to the reviewers below.

Reviewer #1 (Remarks to the Author):

Wilson and co-authors studied GABA_BR action on human and rat neocortical network using electrophysiological methods (whole-cell patch-clamp and local field potentials in vitro. They showed some similarities and differences between rats and human in terms of the modulation of the neocortex through GABA_BRs. They claim that the strength of postsynaptic GABA_BR inhibition declines over lifespan in rodents but this inhibition is stable in human over life. The effect of presynaptic GABA_BRs is more profound in human than rats. GABA_BRs control network oscillations in human. Finally, they showed that the application of anti-seizure-medication levetiracetam for a few days before the tumor surgery enhances GABA_BR inhibition in human.

These data are very valuable taking into account that the study has been done in human tissue. The manuscript should be published after revision.

We thank the reviewer for their positive appraisal of our data. In fact, in collecting further data and re-performing analysis, we now find a similar trend in human L2/3 neurons, as for rats, but over a much longer period. This has been discussed as part of changes to many of the figures and the interpretation.

Major:

-lines 30-33: In the introduction, authors should mention that % of inhibitory neurons in the neocortex is quite different in human and rats/rodents.

We have now clarified that the number of interneurons in human neocortex likely is higher than for rodents – lines 30 - 32

-Figure 1: How was done statistical analysis in Fig1 B? There are different numbers of cells over B-C and D. Please revise it.

We now have revised all statistical analysis. With specific reference to Figure 1B, we used 2-way ANOVA, which is now included in the figure legend, with statistical test results indicated on the figure. The difference in numbers has been corrected.

-Please explain in the methods what “neuronal gain” means and how it was calculated.

Neuronal gain has been described as the relationship of current and action potential output (doi: 10.1523/JNEUROSCI.2045-12.2012) – in essence the slope of the linear phase of the I/F relationship. We measured the FI-slope, as such we have re-labelled the axis labels.

-What is a schematic to indicate number of cells/animals/human individuals? Please provide it to every experiment and figure.

We have now clarified the number of cells/individuals for all datasets, which are given in the figure legend and the number of cells contributing to each dataset given below the graph.

-lines 144-45: "In human cortical neurons, we found that L5 PCs had slightly (+18%) higher baclofen current density, than L2/3 PCs ($t(28,10)=0.69$, $p=0.49$, Holm-Sidak test)."

Is this statistically different? Of not, maybe it should not be considered. Please explain it.

This has now been revised to reflect the new statistical analyses and revised dataset– see lines 130-150.

-Figure 2: please add some examples of the recordings for Fig.A.

The plots shown in Figure 2A are measured holding current, recorded at 20 second intervals for each cell over 15-20 minutes, and then control subtracted. Short of plotting each episodic trace (which should not be concatenated) this would not add further information that which is already presented.

-Lines 193-5: "The decay time-constants of evoked slow-IPSCs were more variable, particularly in adult rats and humans, with a tendency to longer decay times in L5 neurons (Figure 3D)."

Is this statistically different? Of not, maybe it should not be considered. Please explain it.

Also, it should be discussed in more details why kinetics might be different.

We have continued to observe subtle differences in slow-IPSC kinetics even under more stringent statistical analysis. One way that altered kinetics could affect the total inhibition would be to increase the charge of IPSCs. As such, we now plot the charge (measured as the integral) of slow IPSCs – which is not sufficient to overcome layer specific differences in IPSC amplitude.

-Figure 5: E and F would benefit from a better figure legend on the plot. It is hard to follow all the configurations of the stimulation. Please add statistics to the panels E and F.

We have now added a schematics to revised Figure 5 and more added further labels to clarify these data.

-Figure 6: add statistics to the panels D and E.

We have now added statistics to the revised figures.

-Line 369: please specify that these experiments were done in human tissue.

We have clarified the human specific nature of this in lines 444-451 (revised manuscript) and added further labels to Figure 7.

-lines 438-444: It seems that something is lacking here, especially because the second sentence is not finished. Please correct it.

This line has now been corrected.

- lines 443-445: “(...) although individuals who had received LEV, but not experienced seizures tended to possess larger GABABR-mediated IPSCs (Figure 8C).”

Is this statistically different? Of not, maybe it should not be considered. Please explain it.

We have now toned down the text, as more stringent statistical analysis did not reveal a robust difference in IPSC amplitude.

-lines 745-758: add the information about holding potential

Further detail on calculation of holding current measurement has been added. See lines 820-826

-Why Holm-Sidak was used?

We have now changed our entire statistical approach, to use a unified statistical model (see revised statistics section). As such, we use the Tukey test following Type III ANOVA following linear mixed effects modelling, where possible.

-Supplementary Table 2 and 3. There is no data about rats. Please provide it here or as a separate Table.

We have now provided comparison tables for L2/3 and L5 cells for humans and all ages of rats (Supplementary Tables 2-3)

-Supplementary figure 2: please provide number of cells/rats.

All numbers of cells and rats have been added to all figure legends throughout the manuscript and supplementary materials.

-Supplementary figure 5: add figure legends directly above of every plot.

In preparing the revised manuscript, we have removed Figure S5 as we felt this did not add further information or provide support for the main manuscript data.

-Supplementary figure 9: add number of individuals and statistics.

These have been added.

-Supplementary Table 3: this table is not cited in the text, please do this. Also, it is unclear what data are here. Is this data pooled from layers 2/3 and 5?. Comparing to Table 2, these data are mostly similar to L2/3 data. However, e.g. there are different values of SEM for input resistance. Maybe this is a mistake.

Apologies to the reviewer, this table was not well described or integrated. This table was comparing the physiology of L2/3 PCs in control patients to those who had experienced seizures and/or been prescribed levetiracetam. We have now updated the table and revised the legend.

Minor:

-line 344: delete “in”

- line: 692: delete "either"
- line 733: add "were" after "measurements"
- lines 771-773: correct the sentence adding were/was

These minor typos have now been corrected.

Reviewer #2 (Remarks to the Author):

This study uses electrophysiological analysis of neurons in neocortical brain slices from rat and freshly resected human brain tissue to investigate the properties of GABAB receptor signaling on synaptic, neuronal and neuronal network function. The data show several findings:

- *there are cell type-specific developmental changes in rat cortical pyramidal neuron GABAB responses.*
- *human neurons appear to have GABAB-mediated responses that are differentially located in the cell bodies of layer 2/3 and layer 5 pyramidal neurons.*
- *pharmacological activation of GABAB receptors can alter neuronal and neuronal network activity patterns.*
- *treatment of patients with the anti-seizure drug Levetiracetam seems to alter the amount of available GABAB-mediated signaling in human layer 2/3 pyramidal neurons.*

Overall, there are some intriguing findings shown in this paper and some novel information, particularly in relation to the rather unique set of human samples. In a way, there are several strands to the data presented – developmental inter-species comparison, mechanisms of how GABAB effects are mediated, clinical significance of GABAB for epilepsy treatments. Each provides some new findings, but each is slightly incomplete. Furthermore, while the experimental quality is itself of a high standard, there are some limitations in the experimental design and therefore the insights we can make, particularly in relation to realistic activation of GABAB receptors and into some of the proposed mechanisms underlying changes in GABAB-mediated effects.

We thank the reviewer for their interest in our study. In preparing the revised manuscript, we hope with the inclusion of new data and more robust statistical analysis that we have allayed their concerns.

Specific points:

As acknowledged in Discussion, the youngest human samples are from ~30 years old, which is probably more equivalent to 6 month rather than 1 month old rat. So age comparisons are not really overlapping. For example, the developmental decrease in GABAB current density in rats is probably too early developmentally to be captured by the human ages sampled. An older rat age, or young human ages, would allow more directly comparisons of age-dependent changes across species. Without this, there are concerns that the inter-species comparisons, that are a key part of the first part of the paper, are not strongly underpinned.

The reviewer's point is pertinent. First a point of clarification, our 6-month group was actually 6-8 months, which is now clarified in the revised manuscript. Based on our current access to human tissue, obtaining neurosurgical samples from younger patients (e.g. paediatric surgeries) is not possible. To address their concerns, we performed recordings from older rats (12-14 month old) which represent the oldest animals we could obtain under our existing rat breeding. Recordings from these animals did not differ statistically from 6-8 month old rats (see revised Figures 1-3). Furthermore, we serendipitously obtained a number of samples from younger human individuals (<35 years). In revising the manuscript, we have now performed parallel analysis in rats and humans, exemplifying age and species effects.

It is important to note that changes in GABAB-driven currents could reflect changes in the downstream potassium channel expression/properties or G-protein signaling in addition or instead of changes in GABAB receptors themselves.

A discussion of the potential role of K channels in driving postsynaptic GABA_BR maturation has been added (Discussion lines 598-600).

Throughout the data analysis, cell or slice is used as the experimental unit. This is likely a reasonable choice but there is clearly great potential, particularly in the human data, that there will be significant effects associated with the individual samples (perhaps due to case-to-case variation in tissue handling, patient demographics or clinical status). The point is made fairly that in the Methods, with low numbers of recordings from each sample, it may be difficult to measure inter-sample variance accurately. However, this information could be included in hierarchical/mixed statistical models. At the least, it is important to be clear how many samples are associated with each dataset (as well as cell/slice n numbers), either in the text or represented in the datapoint symbols.

The reviewer is correct that we wanted to capture as much cell-to-cell variability as possible, thus report mean and variance from cell averages. To address your valid concerns about variance and potential pseudoreplication, we have now performed linear mixed effects modelling (where possible) in the revised manuscript – accounting for random effects, such as: patient, experimental day, slice. This revised statistical approach is summarised in the methods and applies to data presented in most figures

Line 234 – it is suggested GABAB receptors preferentially localize to the cell soma with age but there is no direct measurement of receptor density there, which is the suggested mechanism. Could a direct measure of receptor density in somatic membranes be made using some immunohistological or electron microscopy approach?

In our revised manuscript, we perform qualitative immunohistochemical labelling for the GABA_{B1} receptor subunit in fixed tissue from the same species and ages as for our physiological recordings (revised Figure 3A-C) which reveals differences in receptor localisation within cortical columns in humans compared to rodents.

Regarding somatic receptor density. GABA_{B1} receptor localisation to the endoplasmic reticulum in cell somata is dictated by expression of the GABA_{B2} receptor subunit to mask the ER retention motif. Thus, somatic labelling likely does not reflect the membrane expression of functional GABA_BRs (Kulik et al., 2002, 2003; Booker et al., 2017). Performing detailed quantitative expression of GABA_BRs at

the cell membrane of identified neurons would require pre-embedding immunogold electron microscopy of cells post recording. While this is an interesting avenue that would add further information regarding compartment specific receptor expression, this is out with the remit of our current study.

Figure 5 – were the input pathways confirmed as independent via interleaved paired pulse stimuli?
Paired stimuli were indeed interleaved. This detail has now been added to the methods.

What is really missing is effect of endogenous GABAB receptor activation on synaptic stimulation. When CGP is added at the end of the baclofen experiments, the EPSC size does not get bigger than baseline, so doesn't this suggest that there is no GABAB-mediated effect triggered by the synaptic stimulation. Maybe this low-frequency, single test pulse is not sufficient to trigger enough GABA release, but it does seem possible with the high-frequency, 5 pulse stimulus train used in Figure 3. It would be good to show that EPSCs evoked by such a high-frequency stimulus train are indeed altered by the presence of CGP – this would confirm an effect of synaptically-evoked GABAB receptor activation.

The purpose of this study primarily was to compare the synapse specific GABA_BR receptor content systematically between synapse groups, which is best achieved with pharmacological activation to remove potential variance due to differential circuit preservation and stimulus variability. However, to address the reviewers justified concern, we have performed a subset of recordings from L2/3 of human neocortex in which we delivered trains of stimuli to recruit volume transmission of GABA and examine short term plasticity before and after CGP application (Revised Figure 5I-K). These data confirm that endogenous GABA release does indeed contribute to presynaptic release in human cortical neurons.

Likewise, in the oscillation experiments shown in Figure 6, would we not expect GABA release to be extensive during the KA/CCh-triggered oscillatory activity, and therefore be activating GABAB receptors. It would be more interesting to see any effect on oscillations of blocking GABAB receptors with CGP rather than activating them all artificially with baclofen.

This is true. We have included new data showing the effect of CGP on oscillations (Figure 6A, B, F).

Line 440 – sentence appears to have been cut off.

This has been corrected.

Lines 442-445 – “We found no overall difference in GABABR-mediated IPSC amplitudes between any patient group (KW=4.47, P=0.107, Kruskal Wallis Test), although individuals who had received LEV, but not experienced seizures tended to possess larger GABABR-mediated IPSCs (Figure 8C).” The assertion about these larger currents seems to correspond to data from two cells (not clear if from the same patient) and is not corroborated by the stats assessment, so should be removed.

We have now refined the statistics and toned down this difference.

How well does 100uM LEV correspond to doses in the brains of patients? This would be discussed.
Based on previous pharmacokinetic data (reviewed in Patsalos 2004), plasma concentrations of LEV

reach ~100 μM rapidly following oral administration of 1000 mg of LEV – the daily dose received by our patients. We cannot guarantee this is the final CSF concentration in humans, but parallel pharmacokinetic analysis at broadly equivalent doses results in a CSF concentration of approximately 100 μM (Doheny et al., 1999). We have added a line noting these details to the results (lines 522-523).

There should be discussion about how to reconcile that LEV patients have greater GABAB currents evoked by baclofen but this same treatment has no differential effect on oscillatory activity.

Further discussion of this point has been added (lines 655- 659)

Wilson et al., Response to Reviewers:

We thank the reviewers for their appraisal of our revised manuscript. We have addressed their specific comments in the revised manuscript text. We provide a point-by-point response below.

Reviewer #1 (Remarks to the Author):

Wilson and co-authors studied GABA_BR functioning in human and rat neocortical networks using electrophysiological methods in acute slices. The experiments are elegant and performed with high standard. Studies on human tissue are always valuable. I recommend the manuscript for publication.

The authors have satisfactorily addressed my concerns.

One minor comment. The sentence should be re-written in lines 449-51: "In human neocortex, because we found low concentrations of baclofen inhibited the strength of oscillations across a broad range of frequencies, likely due to the activation of presynaptic GABA_BRs."

This sentence is now corrected.

Reviewer #2 (Remarks to the Author):

The authors have made an impressive response to my queries on the original manuscript. I am satisfied that they have addressed my requests and feel the manuscript is improved. As such, I think the manuscript should be published.

One additional query regarding the immunofluorescence data now shown in Figure 3A - in the representative images, the staining pattern in human tissue appears to be more heavily based on signal from cell bodies, compared to the rat tissue, which has a stronger neuropil element. If this is truly representative of the data as a whole, it would be worth adding discussion of how this might relate to the laminar profiles and the somatic current density differences.

The reviewer is makes a pertinent observation. We have added further discussion of this to results (lines XX) and discussion (lines XX) text.